# ABC transporters are billion-year-old Maxwell Demons

Solange Flatt [1✉], Daniel Maria Busiello [1,2✉], Stefano Zamuner[1] & Paolo De Los Rios [1,3✉]

ATP-Binding Cassette (ABC) transporters are a broad family of biological machines, found in most prokaryotic and eukaryotic cells, performing the crucial import or export of substrates through both plasma and organellar membranes, and maintaining a steady concentration gradient driven by ATP hydrolysis. Building upon the present biophysical and biochemical characterization of ABC transporters, we propose here a model whose solution reveals that these machines are an exact molecular realization of the autonomous Maxwell Demon, a century-old abstract device that uses an energy source to drive systems away from thermodynamic equilibrium. In particular, the Maxwell Demon does not perform any direct mechanical work on the system, but simply selects which spontaneous processes to allow and which ones to forbid based on information that it collects and processes. In its autonomous version, the measurement device is embedded in the system itself. In the molecular model introduced here, the different operations that characterize Maxwell Demons (measurement, feedback, resetting) are features that emerge from the biochemical and structural properties of ABC transporters, revealing the crucial role of allostery to process information. Our framework allows us to develop an explicit bridge between the molecular-level description and the higher-level language of information theory for ABC transporters.

[1] Institute of Physics, School of Basic Sciences, École Polytechnique Fédérale de Lausanne—EPFL, Lausanne 1015, Switzerland. [2] Max Planck Institute for the Physics of Complex Systems, Dresden 01187, Germany. [3] Institute of Bioengineering, School of Life Sciences, École Polytechnique Fédérale de Lausanne—EPFL, Lausanne 1015, Switzerland. ✉email: solange.flatt@epfl.ch; busiello@pks.mpg.de; paolo.delosrios@epfl.ch

Transport processes across membranes are crucial in every living organism[1]: the ability to import or export substrates is essential, for example, to absorb nutrients or to expel metabolic waste, toxins, or drugs. ATP-binding cassette (ABC) transporters represent a very broad family of transporters that are found in most prokaryotic and eukaryotic cells[2,3]. They are transmembrane proteins whose main function is the active, i.e., driven by ATP (adenosine triphosphate) hydrolysis, import or export of selected substrates both through the plasma and through organellar membranes (e.g., between cytosol and endoplasmic reticulum). The major distinction in the ABC family is between importers, that allow the intake of molecules from the environment into the cell, and exporters, which transport molecules in the reverse direction. The dependence of ABC transporters on ATP as an energy source does not come as a surprise, because transport often takes place against concentration gradients, thus against equilibrium thermodynamics, and energy must be invested to support a steady substrate concentration difference across the membrane. Overall, the functional cycle of ABC transporters requires them to be in a conformation suitable to bind substrates on one side of the membrane, then switching conformation upon substrate binding that stimulates ATP hydrolysis, and, after substrate release and nucleotide exchange, switching back to the previous conformation, ready to start a new transport cycle (Fig. 1a)[3]. Although ABC transporters have been extensively characterized both structurally and biochemically[4], a comprehensive framework that integrates the available information into a simple, instructive and physically consistent model is still lacking.

Because of their ability to maintain a concentration difference between the two sides of a membrane, ABC transporters are reminiscent of Maxwell Demons[5]. The Maxwell Demon is an idealized agent (device) that was proposed originally as a challenge to the second law of thermodynamics. In a nutshell, a box is divided in two halves separated by a wall with a door, and is filled with molecules (see Fig. 1b). If the door is open, the molecules will eventually reach their equilibrium state and distribute evenly in the two halves. The Demon operates (opens or closes) the door, and it does so depending on the presence of approaching molecules. If it detects a molecule coming from, say, the left side, it opens the door and let it pass. In the absence of molecules, or if molecules arrive to the door from the right, the Demon will keep the door shut. Repeating this action over and over again, the final result is, intuitively, an accumulation of molecules in the right half of the box, thus establishing a concentration gradient. Since a steady concentration gradient is at odds with thermodynamic equilibrium, it can be sustained only by the constant investment of energy by the Demon. Yet, the Demon does not perform any direct work on the molecules, for example, by forcefully dragging them from one side to the other of the door. It only selects which crossings of the door are allowed, and which are not, provided that they are driven solely by thermal motion (this is also usually called a Brownian ratchet[6]). The solution to this paradox, namely the persistence of a steady non-equilibrium setting without any apparent energy consumption, lies in realizing that the Maxwell Demon is actually an information-processing device going through three basic steps[7]. First, it must collect information (measurement: is there a particle coming from the correct side?). The result of the measurement triggers the feedback, which consists in writing the measurement outcome in physical memory (say, one bit: 0=molecule absent, 1=molecule present), to be then used to decide the state of the door (0=keep the door shut, 1=open the door); eventually, the memory and the door must be reset to allow the Demon to go through a new cycle (resetting: close the door and set the bit to 0). This causal, directed sequence of information-processing steps requires energy because it would

be otherwise incompatible with equilibrium reversibility. Indeed, according to Landauer's principle, the minimal energy budget required for a Maxwell Demon to work is associated with bit erasure[8]. Recently, scenarios where the measurement device is embedded into the measured system itself led to the development of a slightly different model, the autonomous Maxwell Demon. In this case, all information-processing steps happen as part of the dynamical evolution of the system, until it reaches a none-quilibrium stationary state. Chemical networks that collect and process information manifestly belong to this framework[9–11], as well as ABC transporters.

Recent developments in the study of ABC transporters shed light on a large variety of their structural properties and associated specific transport mechanisms. The main categories of transport functions exhibited by ABC transporters are exporters and importers (as well as, although less common, extractors and mechanotransmitters)[12]. Whereas the Nucleotide-Binding Domain (NBD) of ABC transporters share highly conserved sequences, the diversity of the Transmembrane Domain (TMD) might be the key for such a variety of different mechanisms[13]. We discuss below the roles of these domains and how they are consistently included in the proposed model. As a cornerstone and starting point of most, if not all, models proposed for ABC transporters is the alternating access model, an allosteric molecular model for membrane pumps, and more generally, for a wide range of active transporters[14]. This general framework applies well to ABC transporters, and thus motivated many different models[4,15,16], the main ones being the alternating catalytic sites model[17], the ATP switch model[3], the constant contact model[18], and the reciprocating twin channel model[19]. These models are based on the coupling between substrate-dependant and nucleotide-dependant conformational changes, resulting in different rates between the possible conformations, thus converging to a directionality of the transport cycle. From a broader perspective, it suggests the existence of a unifying framework for importers and exporters, whose thermodynamically consistent formulation is still lacking[20,21]. Therefore, it is interesting to focus on a more general picture in which a unifying model grasps the essential features of ABC transporters as a consequence of a generally applicable mechanism. This quest of a unified model to describe ABC transporters is further supported by the minimal kinetic model[15] which showed how it is possible to predict and reproduce experimental phenomenology (either for importer or exporter), such as the stimulation of ATPase activity through substrate binding in the case of importers. However, in this model with only four states, the function of the transporter (as an exporter or an importer) is the consequence of internal properties such as the different transition rates of each conformations, depending on both states of NBD and SBD .

In our work, we go further in the scope of unifying the mechanisms of import and export, by presenting a model in which different states with a same nucleotide coexist, independently of the transport function, exhibiting different transition rates and in particular hydrolysis rates. We thus extend the model so that the directionality of the transport is the consequence of the modulation of the equilibrium constant between these different conformations upon binding/unbinding of the substrate on the SBD. Our model is based solely on present available structural and biochemical information. Its solution reveals that the biochemical conditions that allow ABC transporters to sustain a concentration gradient against equilibrium exactly match the information-processing steps highlighted above. Thus, ABC transporters are autonomous Maxwell Demons that emerged from billions of years of molecular evolution. This formal correspondence further allowed us to quantify the information stored and erased during one full cycle of operations

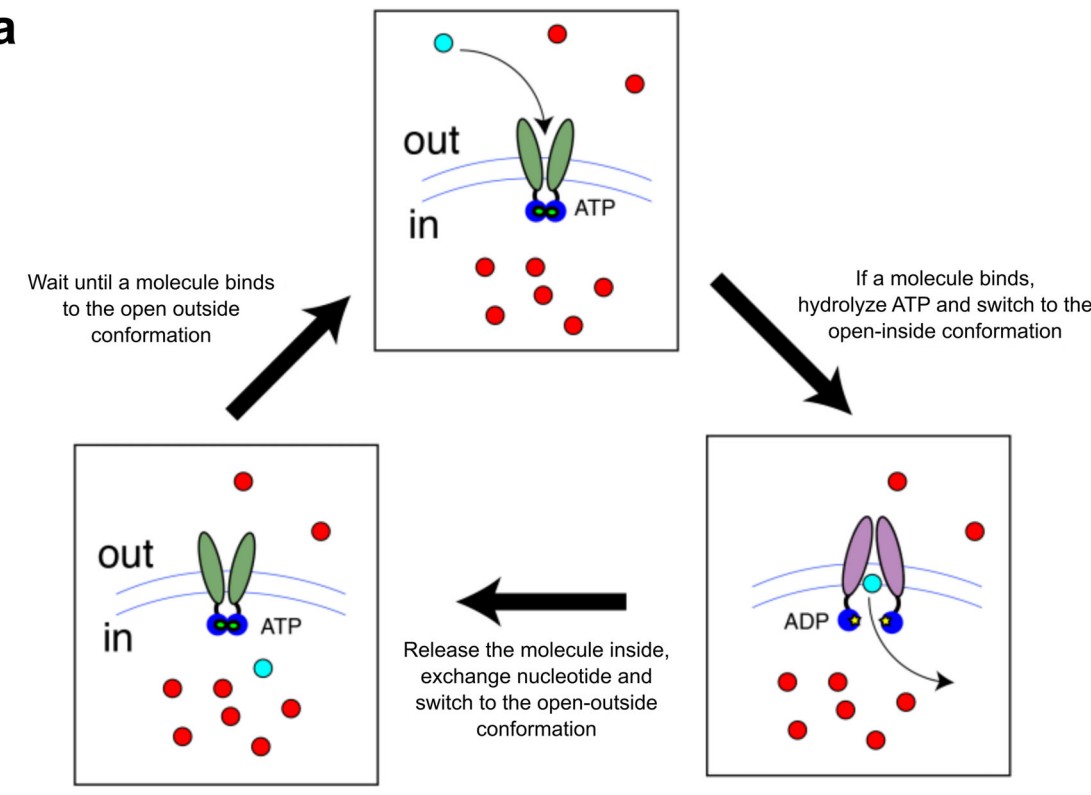

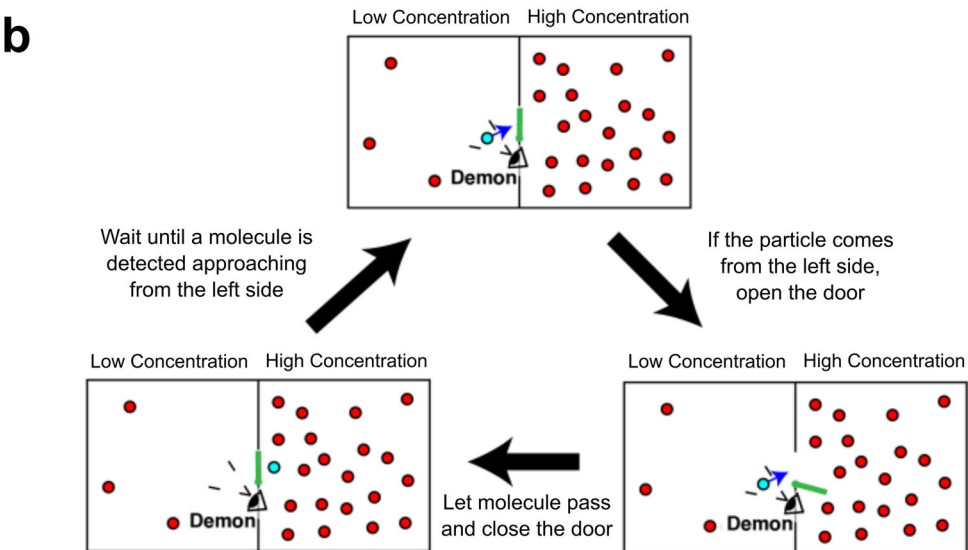

**Fig. 1 ABC transporters and Maxwell Demons. a** In a simplified description, ABC transporters (here an importer) binds a given substrate from the *out* side of the membrane (highlighted in cyan) among the available ones (red circles) in an open-outside conformation bound to ATP (green ovals represent Transmembrane Domains, which are connected to Nucleotide-Binding Domains in blue; ATP is in light green). Upon substrate binding and subsequent ATP hydrolysis, the ABC switches to an open-inside conformation (Transmembrane Domains are in purple, and ADP is indicated by yellow stars). After substrate release in the *in* side and nucleotide exchange, the transporter switches back to the open-outside conformation ready for a new cycle. **b** Among the available molecules (red circles), the Maxwell Demon detects the one approaching the door from the left (in cyan). Then, it opens the door and let the molecule pass, closing the door afterward, and waiting for a new particle to approach.

(measurement–feedback–resetting), showing that the total processed information is, in fact, intimately connected to the steady concentration gradient across the membrane. This approach highlights the role of information processing by ABC transporters, and from a more general perspective, reveals that the high-level information-theory description of biological processes is an emergent property of their detailed nonequilibrium molecular description.

## Results

**A structurally and biochemically based model of ABC transporters.** Structurally, ABC Transporters are dimers constituted by two (usually) identical monomers, each comprising two main subunits: a Transmembrane Domain (TMD) and a Nucleotide-Binding Domain (NBD)[4,12]. The TMDs (two in the dimer) span the membrane and create the channel for the translocation of substrates[3]. The NBDs are located inside the cell, and constitute the binding site of the nucleotides, either ATP or ADP (adenosine diphosphate). For the sake of simplicity, we are going to refer to the membrane side where the NBDs are located as in (or inside), and the side of the membrane where the TMDs protrude into as out (or outside)[3].

Hydrolysis of ATP and nucleotide exchange drive conformational changes that switch the arrangement of the substrate binding site on the two sides of the membrane. When bound to ATP, the TMDs are ready to catch substrates from, or release them to, the out side of the membrane. This conformation is usually called open-outside. Conversely, when bound to ADP the TMDs are open-inside, exchanging substrates with the in side of the membrane. The presence of a bound substrate has a strong impact on ATP hydrolysis or nucleotide exchange rates (or both) typically resulting in an acceleration of the ATPase cycle of ABC transporters[15,22–24]. Since the bound substrate does not directly contact the NBD, it must affect nucleotide processing (hydrolysis and/or exchange) through long-distance, allosteric conformational changes[25,26]. Thus, the ATP-bound state must by necessity be an ensemble of at least two different conformations, characterized by different nucleotide-processing rates, and whose relative equilibrium is tuned by substrate binding so that it can modulate the observed ATPase rate.

We capture these basic features through the model depicted in Fig. 2a. The open-outside, ATP-bound state is split in two states: a slow-hydrolyzing/exchanging state T (TS when bound to substrate), and a fast-hydrolyzing/exchanging state T* (T*S when bound to substrate). The transition rates between T and T* (and also between TS and T*S), and hence their relative populations, can be modulated by the presence of a bound substrate. Indeed, the latter favors the fast nucleotide-processing state, while the slow processing one is preferred when no substrate is bound. This mechanism allows for control of the overall ATPase rate, as observed in experiments, through an indirect, allosteric switch operated by the substrate. For the sake of simplicity, the model proposed here considers that one single nucleotide affects the conformation of the transporter, although a more realistic description would capture the binding of two nucleotides, possibly in their various ATP/ADP combinations.

Upon hydrolysis or exchange, the ATP-bound open-outside states can convert into the ADP-bound open-inside conformation (D without, and DS with, bound substrate). Substrates can bind to the T and D, and unbind from the TS and DS, states. We simplify here the model by assuming that the T* state cannot directly bind substrates from solution, nor the T*S state can release them. Some special care must be paid to the transitions between states bound to different nucleotides, and to the associated rates (red arrows in Fig. 2a). These transitions can proceed via two different routes: exchange or hydrolysis/synthesis (Fig. 2b, green and blue arrows, respectively). The rates of each individual route are related by relations imposed by thermodynamic equilibrium (see Supplementary Note 1 for a detailed derivation). In the absence of a bound substrate:

$$\frac{k_{\mathrm{h}}}{k_{\mathrm{s}}} = \frac{K_{\mathrm{d,ATP}}}{K_{\mathrm{d,ADP}}} \frac{[\mathrm{ADP}]_{\mathrm{eq}}}{[\mathrm{ATP}]_{\mathrm{eq}}} \tag{1a}$$

$$\frac{k_{\mathrm{ex}}^{\mathrm{T \to D}}}{k_{\mathrm{ex}}^{\mathrm{D \to T}}} = \frac{K_{\mathrm{d,ATP}}}{K_{\mathrm{d,ADP}}} \frac{[\mathrm{ADP}]}{[\mathrm{ATP}]} \tag{1b}$$

where $k_{\mathrm{h}}$ and $k_{\mathrm{s}}$ are the hydrolysis and synthesis rates, respectively, $k_{\mathrm{ex}}^{\mathrm{T \to D}}$ is the exchange rate for the release of ATP and binding of ADP, and $k_{\mathrm{ex}}^{\mathrm{D \to T}}$ is the rate of the reverse exchange transition[27]. $K_{\mathrm{d,ATP}}$ and $K_{\mathrm{d,ADP}}$ are the dissociation constants of ATP and ADP, respectively, defined as the ratio between the rates of unbinding ($k_{\mathrm{-T,D}}$) and binding ($k_{\mathrm{+T,D}}$) of the corresponding nucleotide. Analogous relations hold in presence of the substrate. Each pair of transitions, namely $\mathrm{T \to D}$, $\mathrm{T^* \to D}$, $\mathrm{TS \to DS}$ and $\mathrm{T^*S \to DS}$, supports its own rates and dissociation constants for the nucleotides, but always bound to each other by Eq. (1a) and (1b). These relations also imply that, away from equilibrium (i.e., $[\mathrm{ATP}]_{\mathrm{eq}}/[\mathrm{ADP}]_{\mathrm{eq}} \neq [\mathrm{ATP}]/[\mathrm{ADP}]$), the two branches of the reaction are not balanced, and there is a net current that, flowing through the whole reaction network, can potentially result in a net current of molecules across the membrane. Hence, the substrate concentration difference across the membrane must be related to the energy available from ATP hydrolysis, $\Delta G$:

$$\Delta G = k_B T \ln\left(\frac{[\mathrm{ATP}]}{[\mathrm{ADP}]} \Big/ \frac{[\mathrm{ATP}]_{\mathrm{eq}}}{[\mathrm{ADP}]_{\mathrm{eq}}}\right), \tag{2}$$

which vanishes at equilibrium. In Eq. (2), we have neglected the contribution from inorganic phosphate $P_i$, assuming that its concentration is much larger than the one of nucleotides, as in the cell[28], and is thus essentially unaltered by hydrolysis and synthesis.

All the rates of the system are additionally connected by further thermodynamic relations[29], ensuring that if $\Delta G$ were zero, the system would satisfy detailed balance (see Supplementary Table 1). Hence, these thermodynamic constraints reduce the number of independent rates. As an important consequence, thermodynamic relations dictate that the binding of the substrate not only affects the ratio of the T and T* populations ($[\mathrm{T^*S}]/[\mathrm{TS}] \neq [\mathrm{T^*}]/[\mathrm{T}]$) as expected, but it must necessarily modify also their ATPase rates.

We stress here that the proposed model has been built upon structural and biochemical insights, without imposing that it has to operate as an autonomous Maxwell Demon. Moreover, since we only employed features that are common to importers and exporters, all the ingredients introduced here constitute the basis of a thermodynamically consistent unified picture of ABC transporters.

**The emerging logic of ABC transporters.** In keeping with the role of ABC transporters in shuttling substrates against a concentration gradient across a membrane, we look at the steady-state ratio between the substrate concentration on the two sides of the membrane, $[in]/[out]$, which is the most natural quantity to describe the functioning of the transporter.

The steady-state solution of the set of rate equations governing the system can be cast in a very instructive form (see "Methods", and Supplementary Notes 2 and 3):

$$\frac{[in]}{[out]} = \frac{[in]_{\mathrm{eq}}}{[out]_{\mathrm{eq}}} \left[ 1 + \left(e^{\Delta G/k_B T} - 1\right) \left(\frac{K_{\mathrm{e}}^{\mathrm{S}}}{K_{\mathrm{e}}} - 1\right) \left(1 - \frac{k_+^{\mathrm{S}} k_{\mathrm{h}}^{\mathrm{*S}} k_{\mathrm{ex}}^{\mathrm{DS \to TS}}}{k_-^{\mathrm{S}} k_{\mathrm{h}}^{\mathrm{S}} k_{\mathrm{ex}}^{\mathrm{DS \to T^*S}}}\right) F_1(\{k\}) \right] \tag{3}$$

where $K_{\mathrm{e}} = k_+/k_-$, and $K_{\mathrm{e}}^{\mathrm{S}} = k_+^{\mathrm{S}}/k_-^{\mathrm{S}}$, i.e., the equilibrium constants between T and T*, and TS and T*S, respectively. Indeed, $k_+$ is the reaction rate from T to T*, $k_+^{\mathrm{S}}$ the one from TS to T*S, and $k_-$, $k_-^{\mathrm{S}}$ their respective reverse rates (see Fig. 2). Here,

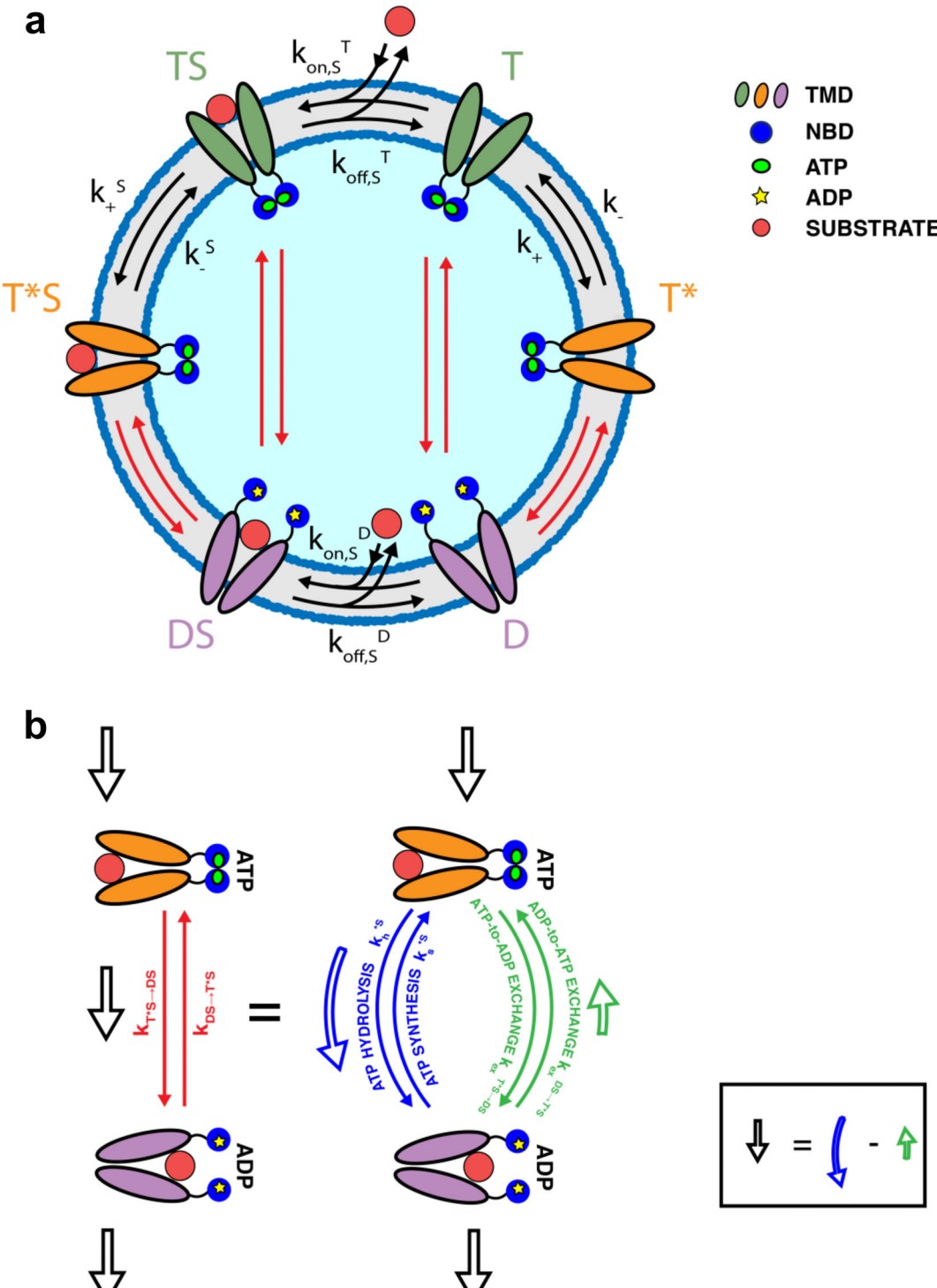

**Fig. 2 Model of ABC transporter. a** ABC transporters are embedded in a membrane, with the nucleotide-binding domains (NBD), indicated by blue circles, sitting on one side of it, and bind ATP (small green circles) or ADP (small yellow stars). Two different ATP-bound, open outside conformations (green or orange transmembrane domains, TMDs) are populated differently depending on the presence/absence of a bound substrate, S (red circles). T and T* indicate these conformations without S, while TS and T*S refer to their counterpart with a bound substrate. The ADP-bound, open-inside conformation (violet transmembrane domains) is accessed from the ATP-bound conformations through composite reactions (red arrows). It is indicated by D when the substrate is not bound, otherwise by DS. **b** The composite reactions are simplified representations of reactions that can take place along different branches: ATP hydrolysis/synthesis (blue arrows) and ATP-to-ADP or ADP-to-ATP exchange; the overall flux through the reaction (black hollow arrow) is the difference between the fluxes taking place along the two branches (blue and green hollow arrows), taken in opposite directions.

$F_1(\{k\})$ is a positive function of the rates, whose specific expression is given in the Supplementary Note 2. The equilibrium concentration gradient is:

$$\frac{[in]_{\text{eq}}}{[out]_{\text{eq}}} = \frac{K_{\text{d,S}}^{\text{D}}}{K_{\text{d,S}}^{\text{T}}} \cdot \frac{K_{\text{d,ADP}} K_{\text{d,ATP}}^{\text{S}}}{K_{\text{d,ADP}}^{\text{S}} K_{\text{d,ATP}}} \qquad (4)$$

where $K_{\text{d,S}}^{\text{D}} = k_{\text{off,S}}^{\text{D}}/k_{\text{on,S}}^{\text{D}}$ is the dissociation constant of the complex of the substrate with the ADP-bound, open-inside conformation, and $K_{\text{d,S}}^{\text{T}} = k_{\text{off,S}}^{\text{T}}/k_{\text{on,S}}^{\text{T}}$ is the dissociation constant of the complex of the substrate with the ATP-bound, open-outside conformation of the transporter. $K_{\text{d,ATP}}$ and $K_{\text{d,ADP}}$ are the dissociation constants of ATP and ADP introduced above for the transporter in the absence of a bound substrate (the same quantities for the substrate-bound transporter are indicated by a superscript S). The specific value of $[in]_{\text{eq}}/[out]_{\text{eq}}$ is dictated by the chemical affinities for the substrates of T and D states (which might be different from each other, in full generality), and satisfies thermodynamic equilibrium, i.e., the products of rates taken forward and backward are equal for every cycle. In Table 1 (see "Methods"), we summarize analytical expressions of dissociation and equilibrium constants.

It is worth highlighting that equations with a structure similar to Eq. (3) have been derived in the context of molecular pumps[30,31], with the aim to encode the effect of chemical kinetics on nonequilibrium steady state. The main difference between the equation presented here and the ones derived in the literature resides in the peculiar dependency on two equilibrium constants, $K_e^{\text{S}}$ and $K_e$[32,33]. We argue that this discrepancy is an intrinsic feature of the chemical architecture of the proposed model, and in particular stems from the presence of the allosteric states T* and T*S. As mentioned, these states are necessary to justify how the substrate can influence the ATPase rate of the transporter by binding to a distal domain. Coarse-graining T and T* together would result in a simpler reaction scheme, with the hydrolysis rates directly depending on the presence of a bound substrate, without an explicit dependence on equilibrium constants. This would yet come at the price of a lack of connection with structural and biochemical features, and to a less clear connection to the crucial ingredients of Maxwell Demons (see below).

Equation (3) reveals the intimate connection between ABC transporters and autonomous Maxwell Demons. In fact, it exhibits the structure of a logical "AND" function for the necessary requirements to let ABC transporters work (each one of them is highlighted by round brackets in the equation for the sake of clarity). Although the crucial roles of nonequilibrium conditions and kinetic effects have already been investigated in molecular machines[30,34] and chemical systems[35,36], here we pinpoint which part of the biochemically informed network is responsible for each information-processing step. In particular, $[in]/[out]$ reduces to the equilibrium value (Eq. (4)) when the following three conditions are not simultaneously satisfied:

i. $\Delta G \neq 0$: energy must be available from ATP hydrolysis. Without an energy source, it is impossible to push molecules against a concentration gradient, and the system falls back to equilibrium.

ii. $K_e^{\text{S}} \neq K_e$: the relative populations of the two ATP-bound states must be tuned by substrate binding. In the absence of the substrate, the transporter preferentially sits in one state (T or T*), and switches to the other in the presence of a substrate (TS or T*S). In the language of the Maxwell Demon, the measurement corresponds to substrate binding, which is followed by the first feedback step, that is the setting of the memory bit by allostery. The necessity for the two equilibrium constants to be different is then clear: if

they were equal, the memory would not change state upon binding, that is, the Demon could not record the measurement outcome, making the subsequent steps immaterial.

iii. $k_+^{\text{S}} k_h^{*\text{S}} k_{\text{ex}}^{\text{DS}\to\text{TS}} \neq k_-^{\text{S}} k_h^{\text{S}} k_{\text{ex}}^{\text{DS}\to\text{T*S}}$: the ratio between these two terms quantifies the weight of the path to import a substrate with respect to the one of exporting a substrate. Transitions between states are always in principle reversible, and this is captured in the model introduced here. As a consequence, importers could in principle work as exporters and vice versa. The inequality between the two rate products ensures that one process dominates over the other by requiring that it proceeds in a definite order: the sequence of events comprising $T \to TS$ (substrate binding, or measurement in the Demon language), $TS \to T^*S$ followed by $T^*S \to DS$ (feedback: writing the outcome of the measurement and opening the door through hydrolysis), and $DS \to D$ followed by $D \to T$, (resetting by substrate release and nucleotide exchange) cannot be equivalent to the opposite one. As a note, the precise import (export) path is immaterial for these considerations, provided the presence of one ATP hydrolysis step and one nucleotide exchange from ADP to ATP. In fact, using the relations enforced by thermodynamic consistency, the ratio between the product of the rates of all these paths is always captured by $k_+^{\text{S}} k_h^{*\text{S}} k_{\text{ex}}^{\text{DS}\to\text{TS}}/k_-^{\text{S}} k_h^{\text{S}} k_{\text{ex}}^{\text{DS}\to\text{T*S}}$ (see Supplementary Note 4).

These three conditions exactly correspond to the three necessary requirements for the successful action of a Maxwell Demon: (1) it must consume energy; (2) the measurement (here, substrate binding) must be recorded on a physical storage device; (3) operating the door must depend on the gathered information, and the system must thus have a preferential direction for biochemical cycles. These three requirements must all be satisfied if the system has to be moved away from equilibrium, and the structure of Eq. (3) precisely captures the logical AND-like condition through a product of three factors, each of which must not vanish.

The solution of a model based only on basic structural and biochemical information has thus revealed that ABC transporters indeed are autonomous Maxwell Demons, with the demon and the door blended into a single physical system[37]). Nevertheless, we have shown that measurement, feedback, and resetting operations have clear biochemical counterparts. Later on, we will use this analogy to build an information-theoretic formulation the model, employing a energy balance for each cycle of operations. The structure of Eq. (3) also reveals that reversing the sign of the triple product turns an importer ($[in]>[out]$) into an exporter ($[in]<[out]$), leaving many routes for nature to evolve one from the other, by tinkering with the transition rates. This observation highlights the fact that the proposed model points toward a unified description of ABC transporters based on the same essential requirements.

**Details of the model tune the performance, while the logic is robust.** The functional form of Eq. (3) states that the logic of the system does not depend on the choice of the parameters (rates) of the system, whose values can only decide how well transport takes place. Thus, to explore their role we numerically solved the model. For the sake of simplicity, we fixed some of the parameters: we chose $k_{\text{off,S}}^{\text{D}} = k_{\text{off,S}}^{\text{T}}$ and $k_{\text{on,S}}^{\text{D}} = k_{\text{on,S}}^{\text{T}}$, so that the substrate dissociation constants did not depend on the nucleotide state of the transporter, $K_{\text{d,S}}^{\text{D}} = K_{\text{d,S}}^{\text{T}}$. Moreover, we also set the nucleotide dissociation constants (both for ATP and ADP) to be independent on the presence or absence of a bound substrate, i.e.,

$K_{d,ADP} = K_{d,ADP}^S$ and $K_{d,ATP} = K_{d,ATP}^S$. With these choices, from Eq. (4), $[in]_{eq}/[out]_{eq} = 1$. Furthermore, we chose $k_{ex}^{T \to D} = k_{ex}^{T^* \to D}$ and $k_{ex}^{D \to T} = k_{ex}^{D \to T^*}$, and analogously for the exchange rates for the substrate-bound transporter. With this simplification and using the thermodynamic constraints on biochemical rates, condition (iii) from the previous section reduced to $k_h^{*S}/k_h^S = k_h^*/k_h \neq 1$. Finally, we fixed the equilibrium constant between ATP and ADP, $[ATP]_{eq}/[ADP]_{eq} = 10^{-9}$, whereas in physiological condition $[ATP]/[ADP] \simeq 10^{28}$ (see "Methods", Table 2, for the values of rates used in simulations).

We thus inspected, one by one, each of the necessary requirements:

i. Access to an energy source, $\Delta G \neq 0$.
   In equilibrium conditions, $\frac{[ATP]}{[ADP]} = \frac{[ATP]_{eq}}{[ADP]_{eq}}$, as predicted by Eq. (3), no transport is possible (Fig. 3a). For large values of the available energy, a plateau emerges, whose presence is due to rate-limiting steps that hinder the ability to fully convert the difference between the excess chemical potentials of ATP and ADP into a chemical potential difference across the membrane. The value of the plateau depends on all the parameters of the system.

ii. Physical underpinning of feedback ($K_e^S \neq K_e$).
   The plateau in Fig. 3a vanishes if $K_e^S = K_e$, as predicted from the analytical solution, and becomes more pronounced as they become more different. This result signals that the transporter can best use the available energy when it can clearly tell the difference between the presence or absence of a substrate or, in a Maxwell Demon language, when the result of the measurement, to be used by the subsequent steps, can be stored more and more unambiguously. It is also noteworthy that as the ratio between the equilibrium constants changes across the unitary value, importers turning into exporters, as predicted in Eq. (3).

iii. Directionality of biochemical cycles ($k_h^{*S}/k_h^S = k_h^*/k_h \neq 1$).
   In typical cellular conditions ($\alpha = 10$), Fig. 3b shows that $\frac{[in]}{[out]} = \frac{[in]_{eq}}{[out]_{eq}}$ whenever $k_h^*/k_h = 1$, irrespective of all rates, as expected. Furthermore, changing the ratio from being larger to smaller than 1, keeping all other rates fixed, changes importers into exporters, and vice versa. Also, as expected, changing at the same time the inequalities for $k_h^*/k_h$ and $K_e^S/K_e$ does not change the transport direction. The unitary value of the hydrolysis rates ratio corresponds thus to the perfect balance between inward and outward active, i.e., energy-consuming, transport. The peculiar

behavior for very large values of the $k_h^*/k_h$ ratio, with the $[in]/[out]$ ratio asymptotically moving back to equilibrium, is a consequence of the very large value of $k_h^*$ (since $k_h$ is fixed for these figures) and of the corresponding synthesis rate (because of Eq. (1a)). Indeed, as $k_h^*$ and $k_s^*$ increase, the flux is more and more carried by the cycle $T \to TS \to T^*S \to DS \to D \to T^* \to T$ always using the hydrolysis/synthesis reactions, which satisfies equilibrium thermodynamic constraints (see S.I., Eq. (S21)).

The emerging logic of the model is actually robust upon the inclusion of further details. For example, we could allow the T* conformation to bind/release substrates, with dissociation constant $K_{d,S}^{T^*}$. Thermodynamic consistency relations would then require $K_e^S/K_e = K_{d,S}^T/K_{d,S}^{T^*}$, giving a clearer interpretation to the molecular stabilization of the T* conformation upon substrate binding. Considering thus also these new binding/unbinding processes led to a dependence of $[in]/[out]$ on the absolute substrate concentrations, which was absent from the simpler model. In particular, substrate excess had here an inhibiting effect (Fig. 4a), as also observed in experiments[38]. Nonetheless, $[in]/[out]$ was shifted from its equilibrium value only if each of the three above conditions was respected, the dependence on the absolute concentrations thus restricted to the functional form of $F_1$ in Eq. (3). Hence, the resulting nonlinear relation between $[in]$ and $[out]$ reproduces a previously unclear observed behavior that, again, relies on the same information-processing requirements detailed above. The robustness of the underlying "AND" structure were to be expected, because, whereas the overall performance of the transporter might depend on the details of the model and on its associated parameters, its internal logic represents a hard set of rules.

The full kinetic reversibility of the present model allowed us to establish a connection between the available energy and transport. Indeed, had synthesis been neglected, the free energy liberated by ATP hydrolysis would have diverged, positioning the system always in the plateau region of Fig. 3a. Considering synthesis explicitly, instead, made it possible to find conditions where there was a net production of ATP from ADP, typically in connection with a substrate flux across the transporter opposite to the one that would have been supported in physiological conditions, thus forcing importers to export molecules and vice versa. For example, this would happen in the presence of an excess of ADP over ATP beyond what their thermodynamic equilibrium would dictate and/or in the presence of a substrate ratio $[in]/[out]$ in excess over the one that the system would have at steady state, given the other conditions (Fig. 4b). This result is consistent with

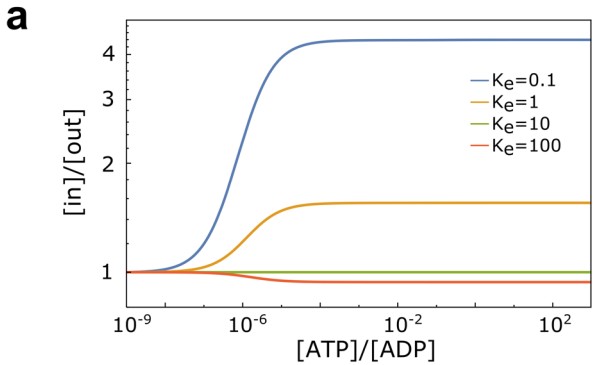
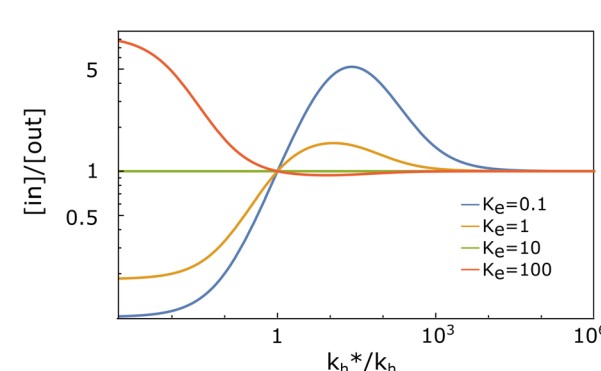

**Fig. 3 Performance of ABC transporters. a** Steady-state concentration ratio $[in]/[out]$ as a function of the ratio $[ATP]/[ADP]$, as a proxy for the available energy in the system. Results are shown for different values of $K_e$, with $k_h^*/k_h = 10$ and $K_e^S = 10$. **b** $[in]/[out]$ as a function of the feedback efficiency on hydrolysis rates $k_h^*/k_h$. It is shown for $K_e^S = 10$ and different values of $K_e$, in a far-from-equilibrium regime, with $[ATP]/[ADP] = 10$ and $[ATP]_{eq}/[ADP]_{eq} = 10^{-9}$, resembling typical cellular conditions.

**a**
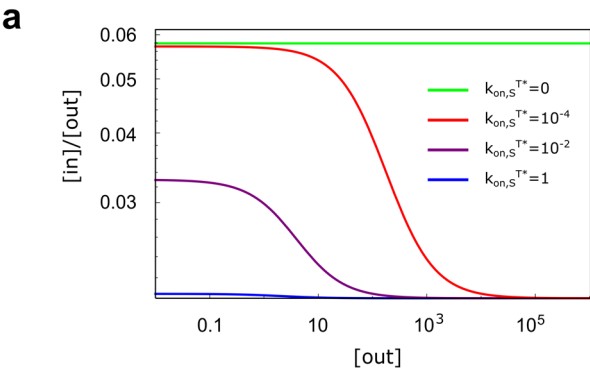

**b**
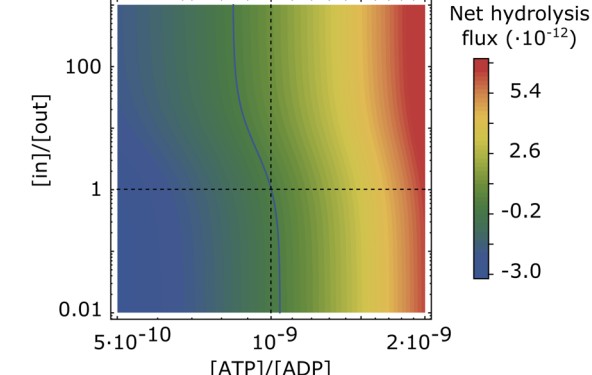

**Fig. 4 Substrate excess has an inhibiting effect. a** Steady-state ratio [in]/[out] as a function of substrate concentration outside the cell for different binding rates $k_{\mathrm{on,S}}^{\mathrm{T*}}$ between T* and T*S. **b** Net hydrolysis rate $\Phi_h$ in a close-to-equilibrium regime, when a substrate concentration gradient is imposed on the system. The horizontal and vertical dashed lines represent equilibrium conditions (respectively [in]/[out] = 1 and $\alpha = \alpha_{\mathrm{eq}} = 10^{-9}$). The continuous line corresponds to the limit when hydrolysis and synthesis fluxes are equal ($\Phi_h = 0$).

the detection of ATP synthesis from ADP by ABC exporters when forced to import substrates in the presence of excess ADP[39]. Thus, having a thermodynamically consistent model is a necessary condition to reproduce the peculiar behavior as an ATP synthesizer.

Summarizing, our description of ABC transporters provides a unique framework encapsulating importers and exporters, identifies the necessary biochemical ingredients to implement information-processing steps, and admits unexpected outcomes that have been experimentally observed but still not satisfactorily explained.

**The cost of processing information.** Besides its paradigmatic simplicity, the Maxwell Demon metaphor bridges between the microscopic, molecular description and the more abstract, but not less fundamental approach of information theory. Indeed, as previously mentioned, the solution to the thermodynamic paradox of the Maxwell Demon operation came precisely by considering the free-energy cost of information processing[8,40,41]. The exact correspondence between ABC transporters and Maxwell Demons in their autonomous version, thus, calls for a more direct approach in terms of information theory.

In particular, we expected that the excess chemical potential across the membrane

$$\Delta\mathcal{E} = k_B T \ln\left(\frac{[in]/[out]}{[in]_{\mathrm{eq}}/[out]_{\mathrm{eq}}}\right), \qquad (5)$$

had to correspond to the available free energy from ATP hydrolysis, $\Delta G$, decreased by the free energy associated to acquiring information (measure), using it (feedback) and erasing it (reset)

$$\Delta\mathcal{E} = \Delta G - T\Delta S_{\mathrm{feedback}} - k_B T(I_{\mathrm{measure}} + I_{\mathrm{reset}}). \qquad (6)$$

$\Delta\mathcal{E}$ is already discounted of the intrinsic chemical potential difference of the substrate across the membrane, and thus applies irrespective of the value of $[in]_{\mathrm{eq}}/[out]_{\mathrm{eq}}$. This is in analogy with Eq. (3), where the part corresponding to the Maxwell Demon multiplies $[in]_{\mathrm{eq}}/[out]_{\mathrm{eq}}$, irrespective of its value.

In the case of an importer, the measurement operation corresponds to substrate binding and to the consequent transition from the T to the TS state. The associated information change is[40]

$$I_{\mathrm{measure}} = \ln\frac{P_3(\mathrm{TS})}{P_3(\mathrm{T})} \qquad (7)$$

where $P_3(\mathrm{T})$ and $P_3(\mathrm{TS})$ are the stationary probabilities of the T

state conditional to the absence or presence of a bound substrate, respectively. They must thus be computed on the corresponding restricted 3-state linear subsystems (T-T*-D and TS-T*S-DS) depicted in Fig. 5 (see "Methods"). A similar argument can be used for $I_{\mathrm{reset}}$ which corresponds to the release of the substrate, $I_{\mathrm{reset}} = \ln P_3(\mathrm{D})/P_3(\mathrm{DS})$.

$\Delta S_{\mathrm{feedback}}$ is the entropy change into the environment given by the actuation of the feedback mechanism. An ABC importer transports substrates across the membrane through ATPase cycles in which both hydrolysis and nucleotide exchange are used (all other cycles always respect equilibrium, according to thermodynamic constraints). In particular, energy is exploited hydrolyzing ATP when the substrate is bound (after measure), and exchanging nucleotide when the substrate is unbound (after reset), so that the system is ready to measure again and restart the cycle. As expected, independently of the specific cycle considered, as long as it runs according to this rule, it dissipates (see Supplementary Note 5)[42]

$$\Delta S_{\mathrm{feedback}} = k_B \ln\left(\frac{k_+^{\mathrm{S}} k_{\mathrm{h}}^{*\mathrm{S}} k_{\mathrm{ex}}^{\mathrm{D}\to\mathrm{T}}}{k_-^{\mathrm{S}} k_s^{*\mathrm{S}} k_{\mathrm{ex}}^{\mathrm{T}\to\mathrm{D}}}\right), \qquad (8)$$

which is precisely the energy dissipated into the environment (that can be converted into heat or entropy difference of chemostatted nucleotides) by opening the door after measure and closing it after the molecule has passed ($k_+^{\mathrm{S}} k_{\mathrm{h}}^{*\mathrm{S}} k_{\mathrm{ex}}^{\mathrm{D}\to\mathrm{T}}$) rather than going through the opposite process ($k_-^{\mathrm{S}} k_s^{*\mathrm{S}} k_{\mathrm{ex}}^{\mathrm{T}\to\mathrm{D}}$).

Equation (6) is an energy balance for a determined cycle of information-processing steps. Indeed, the information associated with the measurement, the dissipation of the feedback, and the resetting contribution are evaluated for one single operation, without averaging over the distribution of all states. The necessity of using this description comes from the fact that ABC transporters are autonomous Maxwell Demons, and an average over the whole system (blended with the measurement device) would cause the impossibility to properly identify the steps that process information. Indeed, this job is performed by the proper splitting into two subsystem, as shown in Fig. 5.

With these ingredients, it is possible to show that Eq. (6) coincides with Eq. (3) (by taking the logarithm of both sides), showing that starting from an information-theoretic approach immediately provides the correct structure of the solution, to be then made explicit using the biochemical rates.

A different information-theoretic approach proposed in the literature, and applied to several molecular machines[11,43–45], emphasizes the identification of terms accounting for information

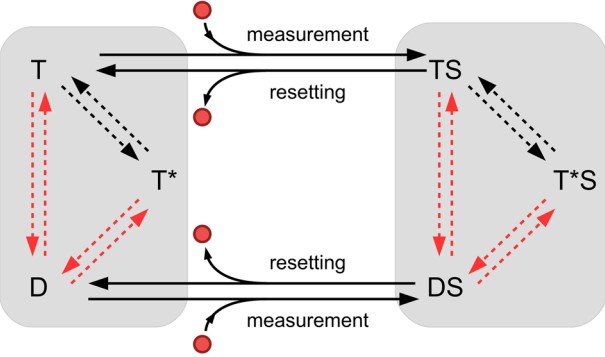

**Fig. 5 ABC transporters as autonomous Maxwell Demons.** Each of the gray rectangles corresponds to a three-state system either with a bound substrate (TS-T*S-DS) or without it (T-T*-D). T and D indicate the conformations in the ATP and ADP state, respectively, while T* is the allosteric conformation. The same terminology holds when a substrate is bound. Solid arrows correspond to measurement and resetting processes, respectively materialized by binding and unbinding of substrate (red cicle). Dashed arrows are the rates realizing the feedback. Dashed red arrows identify composite reactions that can take place along two pathways: hydrolysis/synthesis and nucleotide exchange.

flows in the expression of the entropy production rate, leading to an accurate estimation of transport efficiency. These studies consider bipartite reaction networks, whether linear[11] or nonlinear[44], i.e., chemical systems in which each state is characterized by different sub-features (T/D and bound/unbound in our case) that change with independent sets of transitions and not simultaneously. In our case, ABC transporters manifestly exhibit this structure, as ATP hydrolysis/synthesis and nucleotide exchange are two pathways that only regulates the switch between open-inside (ADP-bound) and open-outside (ATP-bound) configurations, while substrate binding/unbinding change the system by modifying its kinetics properties (see also Fig. 5). As such, the proposed model could be investigated using frameworks developed in the literature, and we leave for future works an in-depth analysis along this line.

However, the energy balance presented in Eq. (6) offers an alternative perspective, which is more focused on energy transduction, showing that it is possible to bridge the frameworks of autonomous and non-autonomous Maxwell Demons considering one single cycle of operations and splitting the system in its version before and after the measurement/resetting, with simple information-theoretic arguments to compute each term.

## Discussion

Transport against a concentration gradient is a crucial process for cells, both to import nutrients, to export metabolic products or other noxious molecules (toxins, antibiotics etc.) and to shuttle molecules across internal cellular membranes. ABC transporters, one of the major classes of proteins that carry out this task, have been fairly well characterized both structurally and biochemically. Here, we proposed a model based on these characterizations, which is thermodynamically consistent, and provides a unified framework to describe both importers and exporters. The picture that has emerged bears tantalizing similarities with the Maxwell Demon, which is an idealized device able to shift the steady state of a system from its equilibrium prescription. In this work, we have built on these sources of information to propose a simple model of ABC transporters whose solution shows that they are not simply analogous to Maxwell Demons: they are autonomous Maxwell Demons. Already billions of years ago, evolution has played with the different conformations and their transition rates

to write in the system the precise necessary conditions for the realization of a Maxwell Demon with a built-in measurement device.

Without any doubt our model, which is on purpose as simple as possible, omits several details of the structure and biochemistry of ABC transporters: two ATP molecules are likely to be hydrolyzed for each transported molecule; the products of ATP hydrolysis are ADP, and inorganic phosphate (Pi), which are then sequentially released leading to further sub-conformations; nucleotide exchange might proceed through a fully apo-state (no nucleotides) or through mixed ATP/ADP-bound states. Nonetheless, our work strongly suggests that, even within a more complex but more realistic scheme, it would still be necessary to identify the basic steps of a Maxwell Demon: measurement, feedback and resetting. Indeed, the logical structure pinpointing the necessary ingredients for the functioning of ABC transporters can be directly derived from an information-theoretic approach.

From a biological perspective, besides the basic essential requirements for the transporter to work, much freedom remains in the choice of the parameter values for its optimization, according to a plethora of criteria, which might not be simply the maximization of the gradient across the membrane. Furthermore, our model shows that there are several different routes to turn an importer into an exporter, leaving much freedom to evolve one into the other.

Our conclusions here have consequences beyond the specific case of ABC transporters and apply to any molecular machine able to bring the system it acts upon into a nonequilibrium steady state. Not only it needs an external energy source. It also needs obligatorily a "measurement device", that is, a way to distinguish between different substrates. In ABC transporters, we have shown that this internal device discriminates through a direct interaction with the substrate that, allosterically, modulates the populations of at least two different conformations, thus writing the measurement result in a sort of molecular information bit. Furthermore, it is mandatory that upon interaction, a specific feedback action takes place, which is possible if, for example, the different conformations selected by the substrate allow the energy-consuming cycle to proceed at different rates. These are features that any structural/biochemical investigation must address, together with careful measurements of the kinetic rates of the system, rather than only of its thermodynamic properties (i.e., equilibrium constants and dissociation rates). It would be intriguing to expand this method beyond this single instance, identifying biochemical equivalents of measurement, feedback, and resetting stages in general models of biological and biochemical machinery functioning as autonomous Maxwell's Demons.

The approach that derives from biochemical and structural attributes emphasizes how the interpretation based on information is an emerging feature of the system. Conversely, beginning at a broader scale with the energy balance method in Eq. (6), which accurately accounts for the thermodynamic cost of processing information, suggests that a more abstract, information-theoretic depiction of biological systems, often advocated as the most appropriate beyond the specific details[46,47] can be successfully employed. This perspective maintains its connection to molecular details, but introduces them only at a later stage. This approach serves as a helpful conceptual framework for understanding the otherwise complex biochemical circuitry of these systems.

## Methods
**Derivation of the stationary concentration gradient**. The system evolves according to a Master Equation. The concentrations of the six possible states (T, T*, D, TS, T*S, and DS) satisfy six coupled rate equations, detailed in Supplementary Notes 1 and 2. In Supplementary Note 1, we also introduce the correct

**Table 1 Analytical expression of dissociation and equilibrium constants.**

| Constant | Expression | Name |
|---|---|---|
| $K_{d,ATP}$ | $k_{-T}/k_{+T}$ | Dissociation cst. of ATP without substrate |
| $K_{d,ADP}$ | $0.5\,k_{-D}/k_{+D}$ | Dissociation cst. of ADP without substrate |
| $K^S_{d,ATP}$ | $k^S_{-T}/k^S_{+T}$ | Dissociation cst. of ATP with substrate |
| $K^S_{d,ADP}$ | $k^S_{-D}/k^S_{+D}$ | Dissociation cst. of ADP with substrate |
| $K^T_{d,S}$ | $k^T_{off,S}/k^S_{on,S}$ | Dissociation cst. of the complex TS |
| $K^D_{d,S}$ | $k^D_{off,S}/k^S_{on,S}$ | Dissociation cst. of the complex DS |
| $K_e$ | $k_+/k_-$ | Equilibrium constant between T and T* |
| $K^S_e$ | $k^S_+/k^S_-$ | Equilibrium constant between TS and T*S |

**Table 2 Numerical values for the rates.**

| Rate | Value | Rate | Value |
|---|---|---|---|
| $k^T_{on,S}$ | $0.5\,\mu M^{-1}\,s^{-1}$ | $k^T_{off,S}$ | $0.01\,s^{-1}$ |
| $k^D_{on,S}$ | $0.5\,\mu M^{-1}\,s^{-1}$ | $k^D_{off,S}$ | $0.01\,s^{-1}$ |
| $k_+$ | $1\,s^{-1}$ | $k^S_+$ | $1\,s^{-1}$ |
| $k_h$ | $0.02\,s^{-1}$ | $k^S_h$ | $0.02\,s^{-1}$ |
| | | $k^S_s$ | $2 \cdot 10^{-9}\,s^{-1}$ |
| $k_{-T}$ | $10^{-4}\,s^{-1}$ | $k^S_{-T}$ | $10^{-4}\,s^{-1}$ |
| $k^*_{-T}$ | $10^{-4}\,s^{-1}$ | $k^{*S}_{-T}$ | $10^{-4}\,s^{-1}$ |
| $k_{+T}$ | $0.5\,s^{-1}$ | $k^S_{+T}$ | $0.5\,s^{-1}$ |
| $k^*_{+T}$ | $0.5\,s^{-1}$ | $k^{*S}_{+T}$ | $0.5\,s^{-1}$ |
| $k_{+D}$ | $0.1\,s^{-1}$ | $k^S_{+D}$ | $0.1\,s^{-1}$ |
| $k^*_{+D}$ | $0.1\,s^{-1}$ | $k^{*S}_{+D}$ | $0.1\,s^{-1}$ |

The rates that are not defined in this table ensue from detailed balance conditions and thermodynamic constraints (see Supplementary Table S1).

expressions for composite reactions. We must add to these six equations the normalization condition, i.e., the conservation of the total concentrations of transporters: $[T] + [T^*] + [D] + [TS] + [T^*S] + [DS] = C_{tot}$. The probability to be in a given state, X, is $P(X) = [X]/C_{tot}$, and its value at stationarity is $P^{st}(X)$. Notice that we only consider the states of the transporter to build probabilities, while substrate concentrations, $[in]$ and $[out]$, converge to a steady state. As a consequence, stationary probabilities can be found using the spanning-tree method and are derived explicitly in Supplementary Note 2.

Moreover, at stationarity, there must be a steady rate of binding and unbinding of substrates, which translates into the absence of a net flux both between T and TS, and between D and DS. This condition is:

$$P^{st}(T)[out]k^T_{on,S} - P^{st}(TS)k^T_{off,S} = 0 \iff P^{st}(D)[in]k^D_{on,S} - P^{st}(DS)k^D_{off,S} = 0 \quad (9)$$

We introduce the following notation:

$$\Lambda_X = \sum_{\gamma \in \Gamma^{\to X}} \prod_{\overrightarrow{l}\,\in\gamma} k_{\overrightarrow{l}} \quad X \in \{T, T^*, D, TS, T^*S, DS\} \quad (10)$$

where $\Gamma$ is the set of all spanning trees directed towards X, $\gamma$ an element of this set, and $k_{\overrightarrow{l}}$ a rate belonging to $\gamma$, with the correct orientation. We finally have the following expression for the stationary concentration gradient[48]:

$$\frac{[in]}{[out]} = \frac{k^T_{on,S}k^D_{off,S}}{k^T_{off,S}k^D_{on,S}} \frac{\Lambda_T\Lambda_{DS}}{\Lambda_{TS}\Lambda_D} \quad (11)$$

This is manifestly equal to Eq. (6), since any ratio $\Lambda(X)/\Lambda(Y) = P_3(X)/P_3(Y)$, where $P_3(X)$ is the stationary probability to be in the state X, evaluated as only the three-state subsystem which X belongs to, namely $(T - T^* - D)$ or $(TS - T^*S - DS)$, does exist. We remark that these 3-state subsystems are linear, as substrate binding/unbinding have been identified as measurement/resetting steps. Eq. (11) can be further manipulated to obtain Eq. (3) of the main text, as discussed in Supplementary Note 3.

**Dissociation and equilibrium constants.** The analytical expression of all dissociation and equilibrium constants used in the main text are reported in Table 1.

**Numerical values.** If not stated otherwise, the numerical values used for the simulations are shown in Table 2.

## Data availability
Data sharing not applicable to this article as no datasets were generated or analyzed during the current study.

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

## Acknowledgements
S.F. and P.D.L.R. thank the Swiss National Science Foundation for support through grant 200020_178763.

## Author contributions
S.F., D.M.B., S.Z., and P.D.L.R. conceived the model. S.F. performed analytical and numerical calculations. S.F., D.M.B., S.Z., and P.D.L.R. wrote and reviewed the manuscript.

## Funding

## Competing interests
The authors declare no competing interests.
