## [Peer review file · Communications Physics]

ABC Transporters are billion-year-old Maxwell DemonsReviewers' comments:

Reviewer #1 (Remarks to the Author):

The paper by Flatt et al. is an elegant description of how molecular machines can be considered Maxwell demons. The authors work out the example of ABC transporters in much detail and argue that the allosteric regulation of substrate transport across the plasma membrane is an evolutionary-built Maxwell demon. The main statement of the paper, although not explicitly claimed, it is apparent: every molecular machine that behaves as a thermal ratchet is a Maxwell demon that operates with some kind of allosteric-like feedback. The statement is obviously strong and not quite accepted in the field. A thermal ratchet that rectifies fluctuations is not necessarily a Maxwell demon and this is a point that the authors should better explain. In the original Maxwell demon paradox, there is an intelligent being that makes measurements and it is small enough not to interfere with the process. No intelligent being is present in the ABC transporters. Is this the main difference between the Maxwell demon and the thermal ratchet? It seems to me that the ABC ratchet with its allosteric mechanism is something more similar to an autonomous Maxwell demon than just the standard Maxwell demon. The authors should better explain this. The paper is definitely interesting and deserves publication after the above and the following minor considerations are resolved.

1. Page 1 bottom, "which random door crossings.." reads weird. Then "...is also sometimes called a Brownian ratchet." I think this is commonly called a Brownian ratchet.

2. top page 2: "...because it would be otherwise incompatible.." The energy requirement is related to bit erasure in the Landauer interpretation. That should be said.

3. Page 2, 2nd paragraph. Acronyms TMD and NBD should be defined here as they appear for the first time. The same at the end of the 2nd paragraph for SBD

4. page 3, k_+^S, k_-^S are not defined

5. In Eq.1 I would also define $K_{d,ATP}$. Is this equal to $[ADP]$

$/[ATP]$?

6. Case ii) on page 4. K_e^S and K_e are not defined (at least >>I could not see them)

7. Besides the table1 I would also add a table for the analytical expressions of the most relevant equilibrium constants K 's

8. Figures 1A,1B, and 2B have text and labels too small and are illegible

Reviewer #2 (Remarks to the Author):

In the manuscript "ABC Transporters are billion-year-old Maxwell Demons" the authors Solange Flatt, Daniel M. Busiello, Stefano Zamuner, and Paolo De Los Rios study the steady state dynamics of a linear chemical reaction network. It is taken to represent a minimal model for the activated transport by an ABC transporter of substrate across a membrane.

Since the model is linear, it can be solved exactly and (normalized) concentrations can be used as probabilities.

The results for the steady state concentrations are correct and the analysis is sound. In fact, it is not new. The conditions enumerated after Eq. (4) have been repeatedly stated, notably in many works by D. Astumian starting in the '90, namely, nonequilibrium and "kinetic asymmetry" (see e.g. <https://pubs.acs.org/doi/pdf/10.1021/acs.accounts.8b00253> for a more recent review). It is surprising that these works are not cited by the authors.

The thermodynamic analysis based on measurement, feedback and reset, which is the key point of the paper, is intriguing. However, the point of the paper is to show that the mathematical model falls into the category of Maxwell demons. Since the model has no implicit or explicit time dependence, it should comply with the definition of autonomous steady-state Maxwell demons. (The analysis that authors provide is instead valid for non-autonomous demons). Therefore, my impression should be based on (mutual) information flows, as defined e.g. in <https://doi.org/10.1103/PhysRevX.4.031015>, and applied to linear chemical ratchets in e.g. <https://doi.org/10.1038/s41557-022-00899-z>.

For example, it seems to me that one important aspect to identify a proper autonomous steady-state Maxwell demon is the fact that the equilibrium rate constant between in and out substrate should be one (otherwise there is as well heat associated to mass transport). This should be discussed possibly as a thermodynamic requirement, not as a mere parameter choice.

My other concern is: What do we learn from such rigorous identification? Arguably, there should be a conceptual interest and a practical one.

The conceptual interest should be to understand the fundamental physical requirements to have a chemical (information) ratchet. This is by now well understood in linear systems, to which the present model belongs. Hence, I fail to see the novelty of the paper from this point of view.

In this respect, the general reader should be made aware of previous results mentioned above (citing the pertinent literature) which are re-derived and re-contextualized here.

The practical interest in principle can be manifold. Unfortunately, the authors do not explain to the reader what we can practically learn from recognizing the (present model of the) ABC transport as a demon. One thing can be the prediction of the thermodynamic efficiency. Can the authors show and comment that? And maybe provide more reasons to justify the benefits of their analysis?

I will be happy to read an improved version of the manuscript that addresses the points above. For the moment I cannot recommend publication of the paper as is.

We would like to thank the anonymous reviewers, as their insightful comments really helped to improve the overall quality of the manuscript. Please, find below a point-by-point response. All changes are highlighted in red in one version of the revised manuscript.

RESPONSE TO REVIEWERS' COMMENTS

Reviewer #1:

The paper by Flatt et al. is an elegant description of how molecular machines can be considered Maxwell demons. The authors work out the example of ABC transporters in much detail and argue that the allosteric regulation of substrate transport across the plasma membrane is an evolutionary-built Maxwell demon. The main statement of the paper, although not explicitly claimed, it is apparent: every molecular machine that behaves as a thermal ratchet is a Maxwell demon that operates with some kind of allosteric-like feedback. The statement is obviously strong and not quite accepted in the field.

We thank the reviewer for finding our paper elegant and for the careful reading. As the reviewer points out, it is evident from our arguments and calculations that every molecular machine that behaves as a thermal ratchet takes information from the environment and converts it into another form of energy, i.e., a chemical potential difference in this case. However, not necessarily this is a Maxwell Demon since the measurement process might be an internal operation (see also the answer below). Moreover, the fact that here the feedback is implemented by allosteric conformational changes is peculiar of this model, and we do not want to convey the message that this statement might contain some sort of universality. In the revised version, we specified that the model presented here has been built solely according to structural and biochemical observations, hence its peculiarities might not be shared by other molecular machines. However, we also clarified that the general behavior as an autonomous Maxwell Demon might be a universal emerging behavior of this class of systems.

A thermal ratchet that rectifies fluctuations is not necessarily a Maxwell demon and this is a point that the authors should better explain. In the original Maxwell demon paradox, there is an intelligent being that makes measurements and it is small enough not to interfere with the process. No intelligent being is present in the ABC transporters. Is this the main difference between the Maxwell demon and the thermal ratchet? It seems to me that the ABC ratchet with its allosteric mechanism is something more similar to an autonomous Maxwell demon than just the standard Maxwell demon. The authors should better explain this.

We thank the reviewer for pointing out this unclear point. Indeed, we agree with them on the fact that ABC transporters are akin to autonomous Maxwell Demon (MD), and this was already stated in the paper, most probably only fleetingly and in an unclear way. In particular, at page 4, we highlight this difference as a technical point, even if the reviewer correctly made us aware of the fact that there is an important difference between MD and autonomous MD. In the revised version, we explained this point from the beginning, highlighting that no idealized external agent is present, and measurement/feedback processes are internal operations. Also, according to the second reviewer, we strengthen the literature on this topic, and include a detailed comment about other information-theoretic approaches to similar problems.

The paper is definitely interesting and deserves publication after the above and the following minor considerations are resolved.

1. Page 1 bottom, "which random door crossings." reads weird. Then "...is also sometimes called a Brownian ratchet." I think this is commonly called a Brownian ratchet.

We are very glad that the reviewer deems our manuscript interesting and worth of publication. We modified the text according to the reviewer's suggestion, and tried to clarify the reported sentence.

2. top page 2: "...because it would be otherwise incompatible.." The energy requirement is related to bit erasure in the Landauer interpretation. That should be said.

Thanks for this comment. We included this specification in the revised version.

3. Page 2, 2nd paragraph. Acronyms TMD and NBD should be defined here as they appear for the first time. The same at the end of the 2nd paragraph for SBD

4. page 3, k_{+S} , k_{-S} are not defined

5. In Eq.1 I would also define $K_{d,ATP}$. Is this equal to $[ADP]/[ATP]$?

6. Case ii) on page 4. K_{eS} and K_e are not defined (at least >>I could not see them)

We thank the reviewer for noticing all these inconsistencies of definitions (points 3, 4, 5, 6 above). We fixed them in the revised version of the manuscript.

In particular, $K_{d,ATP}$ is the dissociation constant for ATP, hence defined as the ratio k_{-T}/k_{+T} . We are sorry for having missed it, and we explicitly included this crucial definition in the revised manuscript.

7. Besides the table1 I would also add a table for the analytical expressions of the most relevant equilibrium constants K 's

We think this can be a good addition to our work in order to improve clarity and reproducibility of the results. We included this table.

8. Figures 1A,1B, and 2B have text and labels too small and are illegible

We thank the reviewer for pointing this out. We increase the size of text and labels in these figures to improve readability.

Reviewer #2 (Remarks to the Author):

In the manuscript "ABC Transporters are billion-year-old Maxwell Demons" the authors Solange Flatt, Daniel M. Busiello, Stefano Zamuner, and Paolo De Los Rios study the steady state dynamics of a linear chemical reaction network. It is taken to represent a minimal model for the activated transport by an ABC transporter of substrate across a membrane.

Since the model is linear, it can be solved exactly and (normalized) concentrations can be used as probabilities.

We thank the reviewer for the careful reading of the manuscript. Considering both the transporter and the substrate, the rate equations are not linear in the sense that [out] and [in] depend on time, converge to the steady-state (they are not chemostatted), and appear multiplied respectively by $P(T)$ and $P(D)$ in the rate equations. We think that the reviewer is referring to the fact that the solution in terms of [in]/[out] does not explicitly depend on substrate concentrations, i.e., [in] is proportional to [out] at stationarity. In one of the manuscript's sections, we actually extended the model including the possibility that also T^* binds [out], as pointed out in the text (page 5), leading to Fig 4A. This is a natural extension that slightly complicates the analytical tractability of the model, but is able to explain the experimental observation that [out] can act as an inhibitor, so that [in]/[out] has an explicit dependence on [out], being in particular proportional to $[out]^2$, hence non-linear. Moreover, we show, although numerically, that the same logical rules hold also in this case, making them robust against non-linear modifications, as logical rules should indeed be. We added some sentences to highlight it in the revised one. It also reinforces the novelty and advantages of the proposed model (see also the answer below). If the referee deems it necessary, we will also add some formulas on the non-linear part.

The results for the steady state concentrations are correct and the analysis is sound. In fact, it is not new. The conditions enumerated after Eq. (4) have been repeatedly stated, notably in many works by D. Astumian starting in the '90, namely, nonequilibrium and "kinetic asymmetry" (see e.g. <https://pubs.acs.org/doi/pdf/10.1021/acs.accounts.8b00253> for a more recent review). It is surprising that these works are not cited by the authors.

We are glad that the reviewer found our results correct and sound. However, we respectfully disagree with the fact that it is not new. Surely enough, solving a rate equation is not new, and the fact that kinetics plays a leading role out-of-equilibrium, on par with energetics, is also known. We completely agree with the referee on the fact that this aspect has to be stressed, and we added some references accordingly. Yet, the model proposed here is new by itself. It has been built solely following structural and biochemical insights, and reproduces several experimental observations concerning ABC transporters for which an explanation was not given before. In particular, we are referring to the inhibiting role of [out] in Fig. 4A and the behavior as an ATP synthesis enzyme in Fig. 4B. At any rate, from reviewer's comment, we realized that some of these claims had been not emphasized enough in the earlier version of the paper. Now, we stress not only on the information-theoretic mapping, but also on the model itself, its novelty, and the fact that it has not been built *a priori* just to work as a Maxwell Demon (MD).

The thermodynamic analysis based on measurement, feedback and reset, which is the key point of the paper, is intriguing. However, the point of the paper is to show that the mathematical

model falls into the category of Maxwell demons. Since the model has no implicit or explicit time dependence, it should comply with the definition of autonomous steady-state Maxwell demons.

We thank the reviewer for deeming our analysis intriguing and for pointing out this missing point. According also to the comments of the first reviewer, we agree that the difference between MD and autonomous MD, although present in the work at page 4, was only fleetingly stated. In the revised version, we stressed that there is no external idealized agent, hence the system has to be interpreted as an autonomous MD.

(The analysis that authors provide is instead valid for non-autonomous demons). Therefore, my impression should be based on (mutual) information flows, as defined e.g. in <https://doi.org/10.1103/PhysRevX.4.031015>, and applied to linear chemical ratchets in e.g. <https://doi.org/10.1038/s41557-022-00899-z>.

We thank the referee for bringing to our attention these two very relevant works. We also found another very interesting work on this subject, and in particular on chemical reaction networks: <https://aip.scitation.org/doi/10.1063/5.0094849>.

In the revised manuscript, we gave credit to these works and their approach. We also would like to point out that $[in]/[out]$ could be also obtained by the time-derivative of the Kullback-Leibler divergence, which relates to the entropy production rate that can be written, in turn, in terms of information flows. In the manuscript, we kept this explanation simple just for the sake of readability.

However, we decided to explicitly present only our formalism for different reasons:

- i) the concepts of measurement and reset can be stated also for an autonomous MD, considering one single cycle of operations. Indeed, the fact that they are autonomous processes effectively introduces some errors due to the stochasticity, and it is also impossible to average independently over the demon and the system, because they are inextricably linked together. The splitting of the system proposed in Fig. 5, along with the energy balance for one single cycle of operations (as quantities are not averaged over any distribution), can then be seen as a way to identify the same steps of a MD in an autonomous MD;
- ii) it gives some additional information with respect to the use of information flows (see also the answers below for details).

In the revised version, we added some sentences on the aforementioned points.

For example, it seems to me that one important aspect to identify a proper autonomous steady-state Maxwell demon is the fact that the equilibrium rate constant between in and out substrate should be one (otherwise there is as well heat associated to mass transport). This should be discussed possibly as a thermodynamic requirement, not as a mere parameter choice.

We thank the reviewer for giving us the opportunity to deepen this aspect of our work. Although we completely agree with the referee that our system falls into the category of autonomous MD, we respectfully disagree with the fact that $[in]$ should be equal to $[out]$. At equilibrium, all cycles of reactions have to be balanced in the Kolmogorov sense, i.e., the product of all rates taken forward is equal to the one taken backward. Given the value of the rates, and then of the chemical affinities for the substrate, the thermodynamic equilibrium does not necessarily state that $[in]=[out]$. As a matter of

fact, the non-equilibrium solution (maintained by a given ratio $[ATP]/[ADP]$), dictates that $[in]/[out]$ is different from its equilibrium value, which is solely defined in terms of rates (Eq. 4), and which is not necessarily equal to 1. Since ABC transporters are biochemical systems, we kept our analysis in full generality, without setting the rates so that $[in]/[out] = 1$, and we do not see a thermodynamic reason to impose this constraint. Although the analytical derivation is general, we imposed some constraints in numerical simulations only for simplicity. Moreover, all the reasonings in the manuscript are formulated in terms of $([in]/[out])/([in]/[out])_{eq}$. We added a sentence in the revised version to better clarify this very important point.

My other concern is: What do we learn from such rigorous identification? Arguably, there should be a conceptual interest and a practical one.

The conceptual interest should be to understand the fundamental physical requirements to have a chemical (information) ratchet.

This is by now well understood in linear systems, to which the present model belongs. Hence, I fail to see the novelty of the paper from this point of view.

In this respect, the general reader should be made aware of previous results mentioned above (citing the pertinent literature) which are re-derived a re-contextualized here.

We definitely agree with the referee on the fact that the message had to better contextualized, hence in the revised version we added some references on this part.

However, the conceptual interest was not to identify the physical requirement to have a chemical ratchet, but to provide a one-to-one mapping between operational steps of an autonomous MD and biochemical/structural elements of an ABC transporters.

Certainly, our interpretation of the steady-state solution retraces the steps of similar works on molecular machines, but the biochemically-informed model we built had no *a priori* purpose to behave as a MD, and converts the aforementioned mapping into a series of practical consequences detailed below. Also, it should come as a pleasant finding that nature has solved and exploited the MD problem billions of years ago!

Additionally, as said above, our approach shows a different procedure to build an energy balance for operational cycles of an autonomous MD.

We added some sentences in the main text to clarify these aspects.

The practical interest in principle can be manifold. Unfortunately, the authors do not explain to the reader what we can practically learn from recognizing the (present model of the) ABC transport as a demon. One thing can be the prediction of the thermodynamic efficiency.

Can the authors show and comment that? And maybe provide more reasons to justify the benefits of their analysis?

We thank the referee for highlighting this missing point. Indeed, we believe that there is a key practical advantage stemming from our work. Whenever a molecule, an ABC transporter in this case, maintains a chemical gradient, experimentalists have to understand how it achieves this effect. For example, allostery might play a leading role in creating faster pathways to favor import or export of molecules, or to consume energy. The detailed analysis proposed here clearly assigns to each transition a specific role in determining the behavior as an information-processing device. From a broader perspective, this message clearly states that the structures of the molecule, the ways the

balance between its different conformations is tilted upon interactions, and the set of reaction rates in molecular transporters are all equally important to fully understand their function.

In the revised version, we added some sentences on this part, to clarify the take-home message of this information-theoretic approach.

Of course, as the referee pointed out, evaluating the efficiency is another important point to characterize molecular transporters. This can be easily achieved implementing the framework proposed in the mentioned work. Since this goes beyond our scope, we decided to add only a comment on this point in the revised manuscript.

I will be happy to read an improved version of the manuscript that addresses the points above. For the moment I cannot recommend publication of the paper as is.

We are happy to resubmit a revised version, hoping that this might meet the criteria of quality and clarity of the journal, along with the referee's appreciation.

Reviewer #1

I am satisfied with the author's response, they properly answered my concerns. I also think they have properly answered the concerns of the second reviewer. His main criticism was about novelty. However, as the authors clarify in their response, this paper shows allostery as a potential mechanism for information transfer at the biological level for the specific case of ABC transporters. This work shows that the operations of measurement, feedback, and resetting in autonomous Maxwell demons might be evolutionary encoded in allostery. This is a beautiful idea that the authors should do good to better emphasize in the abstract where allostery is not ever mentioned. I am happy to recommend the paper for publication.

We thank the reviewer for their positive evaluation of our work, for deeming it worth of publication, and for commenting about our answer to the criticisms of Reviewer #2. As they highlight the beauty of the role of allostery that emerges from our study, we followed their suggestion and emphasize this point in the abstract of the revised version.

Reviewer #2

I thank the authors for trying to address my points and to improve the papers (according to the comments of both referees). However, my doubts and concerns were not dispelled.

In the following I am going to report my original points (with typos corrected), the author's replies, and my present criticisms. I quote only the parts of interest only for brevity—there is no intention to alter the meaning of sentences by taking them out of context. I consider answered all the points I do not discuss. In this respect, I appreciate the effort the authors have put into improving the discussions in the paper.

We thank the reviewer for their detailed report and for appreciating some of the improvements of the revised version. Below, we present a point-by-point answer to all the concerns.

1)

Referee 2: "...Since the model is linear, it can be solved exactly and (normalized) concentrations can be used as probabilities."

Authors: "...Considering both the transporter and the substrate, the rate equations are not linear in the sense that [out] and [in] depend on time, converge to the steady-state (they are not chemostatted), and appear multiplied respectively by $P(T)$ and $P(D)$ in the rate equations..."

Comments:

In fact, I have to rectify my initial statements.

a) After checking more carefully the supplementary informations, I have realized that (S12) are the dynamical equations for [in] and [out], in their stationary state. So, as the authors say, the substrate is not

chemostatted, and the dynamics is nonlinear since there are bimolecular reactions involving the dynamical species.

b) This is however a problem: in general, only for unimolecular (pseudo first order) chemical reactions we can treat probabilities as concentrations. For nonlinear dynamics there is no general correspondence between the two (only in the limit of infinite copy numbers the concentrations are the most likely values). See for instance doi.org/10.1063/1.4935064 and references therein.

As a result, the standard analysis of Maxwell demon with feedback and measure is not well grounded, because it needs to be based on information theory, which clearly need probabilities. For example, Eq. (7) might be reminiscent of an (unaveraged) mutual information, but this observation does not go beyond the level of an analogy.

The new reference [41] that the authors have added goes exactly in this direction: it extends for nonlinear systems the standard information-theoretic analysis of autonomous Maxwell demons (in terms of information flows between a bipartite system) to a description based on concentrations. I am not aware of a similar extension (from probabilities to concentrations) applicable to the analysis made by author, namely, in terms of measurement of feedback. Whether the theory of reference [41] can be used to put on solid ground the analysis the authors are doing here, is yet to be seen. This would constitute a theoretical achievement worth publishing, which I would be delighted to read.

For these reason I tend to believe that the papers does not provide strong evidences for the conclusions of the section "The cost of processing information"

We thank the reviewer for detailing better a point that was unclear to us in their first round of review. Here, we answer to both points a) and b) together, as they are intimately linked.

First of all, it remains unclear to us what the reviewer means by “dynamical equations for [in] and [out] in their stationary state”. If we understood correctly, they intend to say that Eq. (S12) determines the steady-state solution of [in] and [out], *i.e.*, the concentrations of the ligands inside and outside the membrane, respectively. As such, at steady state, [in] and [out] are fixed to their stationary values and the system can be solved for all the remaining states using the Hill’s spanning-tree method. Moreover, since we are studying only a single ABC transporter, we can always name $P(T)$ as the probability of finding it in the state T , and the same holds for all the other states. Indeed, we never wrote $P(\text{in})$ or $P(\text{out})$, since the system is nonlinear and then it cannot be mapped into a Master Equation system, as the referee correctly points out. Hence, we agree with the referee and we hope that this explanation clarifies what we did. We add some specifications along this line in the Methods and Supplementary Information.

Moreover, the reviewer also refers to the information-theoretic approach. Apart from the fact that our framework is correct on a general level (in the sense explained above), in the information-theoretic section we split the system into two linear subsystems, as the nonlinear reactions only enters as measurement and resetting operations. This is a further reason why we are allowed to use $P(T)$, $P(TS)$, $P(D)$, and $P(DS)$ throughout the entire manuscript. Methods now contains some details also about this point.

2)

Referee 2: "The results for the steady state concentrations are correct and the analysis is sound. In fact, it is not new..."

Authors: "We are glad that the reviewer found our results correct and sound. However, we respectfully disagree with the fact that it is not new. Surely enough, solving a rate equation is not new, and the fact that kinetics plays a leading role out-of-equilibrium, on par with energetics, is also known. We completely agree with the referee on the fact that this aspect has to be stressed, and we added some references accordingly. Yet, the model proposed here is new by itself..."

My statement is about the solution of this class of model systems and the identification of the kinetic and thermodynamic conditions leading to nonequilibrium concentrations. The lack of novelty here is accepted by the authors, in their response.

I did not question the novelty in the construction of such model for ABC transporter.

To my eyes this means that the authors provide novel results of interest to (bio)chemists but not really to physicists.

In full honesty, we do not know how to comment this point by the reviewer. Of course, the novelty about the solution of this class of model is not new, and they seem to remark again this point. This would mean that any work solving Master Equation systems is not a novel work. Here, the novelty regards the construction of the model and the interpretation of chemical reactions as operational steps of a Maxwell Demon. We detailed the advantages of our results, as also admitted by the reviewer that appreciated the revised version of the discussions, and we are glad of this.

The implicit statement behind this comment, that physics should only be interested in first principles and not in how they manifest in real systems (biological, in this case) is a respectable opinion of the reviewer, that yet we do not share, and we believe that it should not be employed as a criticism. Indeed, physical chemistry and biophysics are by all means branches of physics.

3)

Referee 2: "Therefore, my impression [is that the analysis] should be based on (mutual) information flows, as defined e.g. in <https://doi.org/10.1103/PhysRevX.4.031015>, and applied to linear chemical ratchets in e.g. <https://doi.org/10.1038/s41557-022-00899-z>.

Authors: "We also would like to point out that [in]/[out] could be also obtained by the time-derivative of the Kullback-Leibler divergence, which relates to the entropy production rate that can be written, in turn, in terms of information flows."

This connects to point 1). If the authors can make this alleged connection with information theory similar to what done in Ref. [41], I urge them to include it in the paper to substantiate their analysis of the second last section.

We thank the referee for letting us comment further about this point. To us, the connection between our work and [41] is evident, but we are happy to detail it better. Our system has the same structure

shown in Figure 1 in [11], which is also the starting setup for [41]. Making explicit reference to [11], the two purple mechanisms are hydrolysis/synthesis and nucleotide exchange, while the teal horizontal arrows are ligand binding/unbinding on T and D side. We also added the presence of the allosteric state, motivated by structural observations, which implements the storage of information by the Maxwell Demon, as detailed in the main text. Moreover, we can further complicate the model, as already discussed in the text, by adding substrate binding/unbinding between allosteric states, resembling the network structure of Fig.2b in [41]. The system is manifestly bipartite. The additional fact that, as said above, we only consider the steady-state solution, with [in] and [out] concentrations equal to their stationary values, allows us to exactly (analytically) map the system into a Master Equation. Hence, the parallel might be closer to [11] than to [41]. Of course, extending our energy balance approach to more complex CRN would be of interested, as also the referee points out later on, and we added this fascinating perspective in the conclusions. In the revised version, we clarify the connection between our work and [11] (or [41]) along these lines.

We mentioned the Kullback-Leibler divergence as the steady-state can be obtained by minimizing the excess entropy production, which is, in fact, the time-derivative of the Kullback-Leibler divergence. Then, rewriting the excess entropy production in terms of the total entropy production, it would be nice to build a connection between the framework of [41] and our work. We agree with the referee that it would be intriguing to draw such a precise link, but this lies beyond the scope of the present manuscript. In order to not lead the reader astray, we decided not to add any discussion about the Kullback-Leibler divergence in the revised version of the manuscript.

We also would like to comment about the necessity of reproducing the steps in [41] (or [11]). The main text is already pretty dense, then introducing novel definition and notation to retrace the steps in [41] (or [11]) is not an optimal choice, in our opinion. Moreover, since the mapping is pretty straightforward, the reader will not learn anything new from it.

We do agree on the fact that it might be interesting to study ABC transporter in terms of information flow. However, the investigation of a core message for this study, apart from a sheer application of a well-known framework, deserves proper attention and a future work on it might be worth.

Authors:"the concepts of measurement and reset can be stated also for an autonomous MD, considering one single cycle of operations. Indeed, the fact that they are autonomous processes effectively introduces some errors due to the stochasticity, and it is also impossible to average independently over the demon and the system, because they are inextricably linked together. The splitting of the system proposed in Fig. 5, along with the energy balance for one single cycle of operations (as quantities are not averaged over any distribution), can then be seen as a way to identify the same steps of a MD in an autonomous MD"

I am sorry but I cannot follow these arguments. What errors are introduced in the stochastic setting? I would kindly ask the authors to provide a thorough set-by-step analysis of the stochastic problem to clarify these statements.

We would like to explain better our argument. The fact that the operations are stochastic introduces some errors that are unavoidable in contrast with the standard Maxwell Demon scenario.

Measurements cannot be perfect and sometimes the autonomous Demon works in the opposite direction with respect to the desired one. This is, of course, intrinsic to the definition of autonomous Maxwell Demon and we did not want to say anything more profound than this.

For what concerns the cycle of operations, since it is impossible to average independently over the demon and the system, we consider all the quantity along a single cycle of operations and construct the energy balance this way.

We are not claiming that this is an approach that can be extended to any system, as we detail also below. We present a way to identify measurement, feedback and resetting in a context of an autonomous Maxwell Demon. This identification comes before the information-theoretic approach (starting from the steady-state solution for [in]/[out]), and hence we believe it is more general than the energy-information balance itself.

As we presented an alternative approach and an identification of operation steps in an autonomous Maxwell Demon, and this is clear from the manuscript, we decided not to add any further clarification. This choice has been also motivated by the fact that the perplexities of the referee concerned our previous response and not the content of the manuscript. However, as said above, we better specified the link with previous information-theoretic approaches (see the point above).

4)

Referee 2: "...one important aspect to identify a proper autonomous steady-state Maxwell demon is the fact that the equilibrium rate constant between in and out substrate should be one (otherwise there is as well heat associated to mass transport)."

Authors: "We respectfully disagree with the fact that [in] should be equal to [out]. At equilibrium, all cycles of reactions have to be balanced in the Kolmogorov sense, i.e., the product of all rates taken forward is equal to the one taken backward. Given the value of the rates, and then of the chemical affinities for the substrate, the thermodynamic equilibrium does not necessarily state that [in]=[out]."

I did not claim that [in] and [out] have to be equal in general, but that [in] should equal [out] in order to respect the usual definition of autonomous Maxwell demons. If [in] and [out] differs the 'measuring+feedback apparatus' and the rest of the system exchanges not only information but also energy. Maxwell demons that have this property are called nonequilibrium Maxwell demon, and have been considered only recently. See for instance doi.org/10.1103/PhysRevE.103.032118 and references therein.

We thank the referee for pointing out the existence of these recent categorizations for Maxwell Demons. However, citing *Phys. Rev. Lett.* 123, 216801 (2019), non-equilibrium Maxwell Demons (or N-demons) are devices in which the system operates by exploiting a non-equilibrium distribution, without acquiring any information through measurement processes on single particles. We are unsure whether the referee refers to this definition having in mind the distribution of ATP/ADP molecules, which is not at equilibrium and provides energy. However, being an autonomous device, categorizing our system as an N-demon because of this sounds a bit strange to us. Anyway, ATP can always be seen as a chemical fuel in a chemical network, without invoking a-priori the existence of a hidden agent that operates on the system. Conversely, we do not see how

$[in]/[out] \neq 1$ at equilibrium is reminiscent of an N-demon. Indeed, this ratio can be different from 1 at equilibrium, and there is no manifest comparison with the definition of an N-demon. At any rate, we believe that it is not necessary to discuss explicitly the characterization of Maxwell Demons, as it does not change or add anything to the message of the work. Moreover, introducing a novel definition can only confuse the reader, in our opinion, as the work is already quite dense of information. We therefore decided to not include the suggested reference or any reference therein regarding the classification of Maxwell Demons. However, if the referee deems this addition crucial to better contextualize the work, we will be happy to add a sentence and the corresponding citations.

5)

Referee 2:"The conceptual interest should be to understand the fundamental physical requirements to have a chemical (information) ratchet.

This is by now well understood in linear systems, to which the present model belongs. Hence, I fail to see the novelty of the paper from this point of view."

Authors:"The conceptual interest was not to identify the physical requirement to have a chemical ratchet, but to provide a one-to-one mapping between operational steps of an autonomous MD and biochemical/structural elements of an ABC transporters...Additionally, as said above, our approach shows a different procedure to build an energy balance for operational cycles of an autonomous MD"

See my comments above and below related to the (biochemical/structural) relevance of the model presented by the authors. Concerning the energy balance/cycle analysis, my other comments apply: I do not see a solid underlying theory that push the conclusions beyond the status of analogies. I really hope the authors can include it in a revised version of the manuscript.

We already commented carefully about the physical and biological relevance of our work, and this question lies more in the realm of philosophy than natural sciences.

About our cycle analysis, the procedure to obtain the energy-information balance seems pretty straightforward to us, given the premises about the identification of probabilities explained above and now clarified in the Methods, thanks to the suggestion of the reviewer.

The core message is the necessity of having measurement, feedback, and resetting steps incorporated in different chemical reactions, and this does not rely on the cycle analysis. The information-theoretic approach, on the contrary, is not applicable to any system, as it is. It would be nice to build a general approach following the steps presented in the main text, and we are working on it, but clearly this should be the content of a different work. We included this perspective in the conclusion.

6)

Referee 2:"One thing can be the prediction of the thermodynamic efficiency. Can the authors show and comment that?"

Authors: "Of course, as the referee pointed out, evaluating the efficiency is another important point to characterize molecular transporters. This can be easily achieved implementing the framework proposed in the mentioned work. Since this goes beyond our scope, we decided to add only a comment on this point in the revised manuscript."

I am very perplexed by the response. If they believe that efficiency is another important point and it can be easily obtained within this framework, why not adding and discussing it to strengthen the content of this manuscript?

As our core message is the necessity of having certain operation steps encoded in the chemistry, and specifically in the allosteric changes, the efficiency is not something we are interested in directly at this stage. Moreover, we do not want to add a definition without having in mind a useful way to implement it in experiments or a path to draw new conclusions about ABC transporters. The reviewer should bear in mind that we are focusing on a specific system, and we believe that the idea is consistent as it is, without the addition of further analysis, just for the mathematical beauty of derive them explicitly.

Referee 2: "...And maybe provide more reasons to justify the benefits of their analysis?"

Authors: "The detailed analysis proposed here clearly assigns to each transition a specific role in determining the behavior as an information-processing device. From a broader perspective, this message clearly states that the structures of the molecule, the ways the balance between its different conformations is tilted upon interactions, and the set of reaction rates in molecular transporters are all equally important to fully understand their function."

This confirms my idea expressed above that the message of the paper (as is now) is best addressed to a more specialized community (biochemistry, structural chemistry, etc.) rather than to the a broad physicists' audience.

We already commented on this point and why we completely disagree with this idea of what physics should care about.

7) Additional comment

Around equations (7) and (8) it is mentioned that entropy is dissipated as heat.

Not all dissipated entropy must go into heat. Certainly, part of it can be freed as enthalpy (heat) but another can go into the entropy difference of the reactants and products that are kept chemostatted (here ATP, ADP and P). See for instance doi.org/10.1103/PhysRevX.6.041064 on page 10.

We thank the referee for this important point. We included this specification in the manuscript.

In view of the above problems that still persist I cannot recommend publication of the manuscript in the present form.

We hope that the reviewer will find the revised version of the manuscript suitable for publication.

Reviewers' comments:

Reviewer #3 (Remarks to the Author):

This manuscript proposes a kinetic model for ABC transporters, a broad family of biological machines performing ATP hydrolysis-driven transport across cells' membranes. The authors choose the ingredients of the model driven by experimental evidence. When analyzed through the lens of information theory, the model shows Maxwell's demon-like features helping its interpretation in terms of information processing and thermodynamics.

A discussion about the novelty of the results presented in this manuscript arose in previous rounds of review. The authors acknowledge in their manuscript that "the crucial roles of non-equilibrium conditions and kinetic effects have been already investigated in molecular machines", including the role of allostery. Also, the fact that dissipative chemical systems can be interpreted as physical realizations of Maxwell's Demons has been shown in the literature cited in the manuscript. As the authors remarked, the novelty of the present contribution should be found in the kinetic model it proposes, which is new for the specific case of ABC transporters, to the best of my knowledge. Such a model differs from other minimalist models proposed in the literature for enzyme-driven transporter and leads to apparently different conclusions about what governs the steady-state substrate concentration on the two sides of a membrane [Acc. Chem. Res. 51, 2653 (2018); J. Am. Chem. Soc. 143, 5569 (2021)]. If the authors clarify why their model differs from other minimalist models, both in the choice of the minimal ingredients and in the predictions (see comment C1 for more details), I would confidently say, to the best of my knowledge, that that constitutes an element of novelty in the kinetic modelling of biologically relevant systems.

Unfortunately, I share some of the doubts of reviewer #2 concerning the rigorousness and generality of the Maxwell Demon analogy.

(i) As the authors notice, "the Maxwell Demon does not perform any direct mechanical work on the system", meaning that the Demon and the system only exchange information. Would Maxwell's Demon analogy hold strictly if the equilibrium ratio $[in]_{eq} / [out]_{eq} \neq 1$? In such a case, the Demon could alter the energy and not only the system's entropy, i.e., the Demon's action raises or lowers the substrate's energy by biasing it towards a region with a different standard chemical potential. In the language of Ref. 11 [Phys. Rev. X 4, 031015 (2014)], this would correspond, I think, to having an energy flow between the two subsystems. In my understanding, an autonomous Maxwell's Demon is such when the Demon (here, all the chemical reactions that do involve the binding and unbinding of the substrate?) and the system (here, the substrate "motion" across the barrier realize by some reactions?) only exchange information.

ii) Did the authors try to check if their definition of Maxwell's Demon is compatible with recent work on Characterizing autonomous Maxwell demons [Phys. Rev. E 103, 032118 (2021)]?

In the two previous paragraphs, I gave my frank opinion about this manuscript in the context of the previous discussion, which I couldn't avoid entering. Overall, my independent assessment of the work is positive, as I find it technically sound and interesting. However, I also think the authors can improve the impact of their work by addressing some "weaknesses" (to be understood as "missed opportunities" to make the paper better in connecting with previous literature). Concretely, I would be pleased to recommend the manuscript for publication after the authors have a chance to consider my two comments below.

STRENGTHS:

S1) The authors did an admirable job going through the biological literature and distilling a new minimal model for ABC transporters, supported by experimental evidence and amenable to analytical analyses. Their model is a significant contribution that I consider worth publishing.

S2) The analysis of the model is insightful. Equation (3) neatly reveals three conditions that must be

contemporarily satisfied to have transport. This theory-driven understanding vouches for the power and usefulness of physical modelling in identifying the fundamental elements of complex phenomena.

S3) The Maxwell's Demon analogy is well-motivated and leads to an elegant interpretation of each factor in equation (3) in terms of information-theoretical concepts. Although the analogy between Maxwell's Demons and chemical systems is not new, the authors' approach looks original in the context of ABC transporters.

WEAKNESSES:

W1) In its current form, the manuscript fails to cite and contrast part of the results with previous similar results obtained in the context of chemical pumping, including other minimal models for enzymatic transporters. I see a missed opportunity to draw an insightful connection here, also in the light of seemingly contradictory results concerning previous models (see comment C1 for more details).

W2) In its current form, the manuscript fails to convincingly demonstrate that the Maxwell's Demon interpretation buys some new practical tools helping to characterize ABC transporters in practice better. This issue was put forward also by Referee #2, and the authors should have replied more convincingly. Let me be provocative: ABC Transporters are billion-year-old Maxwell Demons, and so what? There's room to address this critique by taking inspiration from previous literature (see comment C2 for more details).

COMMENTS:

C1)

Molecular pumps have been analyzed by the molecular machines community, including kinetic models of ATP hydrolysis-driven transporters [Acc. Chem. Res. 2018, 51, 2653–2661; J. Am. Chem. Soc. 143, 5569 (2021)]. Equation (3) is very similar to the typical "pumping equations" derived in the context of molecular machines and motors following the same kind of analysis the authors perform (i.e., using Hill's spanning-tree method). In the context of molecular machines and motors, the analogue of the quantity in brackets in equation (3) is usually called the "kinetic asymmetry factor" and found to not depend on equilibrium constants but only on differences in transition states' energies [Chem. Soc. Rev., 2017, 46, 5491; Chem 6, 1952 (2020)]. The fact that, in the authors' model, the quantity in brackets in equation (3) does depend on (some) equilibrium constants is quite interesting, in my opinion, as it hints at practical ways to determine which model works best for a specific experiment. Therefore, I urge the authors to connect their ABC transporters with previous chemical pumping literature. In particular, it would be valuable to understand whether the dependency on some equilibrium constants in equation (3) is compatible with previous results (and arises, for instance, from model features) or if the authors think their result is somewhat in contrast with previous ones.

C2)

Information-theoretical treatments of biologically relevant systems are typically used to define some performance quantifiers [for a review: Front. Phys. 11:1108357 (2023) doi: 10.3389/fphy.2023.1108357]. What are the authors gaining from calling the terms in equation (3) powering, measurement, feedback, and resetting other than a nice analogy? This question should and can be answered more convincingly by the authors, at least by acknowledging some examples where information-theoretical treatments have been used to quantify performance in biological systems [Pnas 119, 208083119 (2022); arXiv:2209.12084] or suggesting their strategy to put their information-theoretical understanding in practice.

Reviewer #3

This manuscript proposes a kinetic model for ABC transporters, a broad family of biological machines performing ATP hydrolysis-driven transport across cells' membranes. The authors choose the ingredients of the model driven by experimental evidence. When analyzed through the lens of information theory, the model shows Maxwell's demon-like features helping its interpretation in terms of information processing and thermodynamics.

We thank the reviewer for taking the time to read and comment our manuscript, as well as our discussion with the previous reviewer. We appreciate the general positive opinion about our work, and we understand the reviewer's concerns.

Below, we address all the comments, and we acknowledge their role in improving both quality and impact of the manuscript. We hope that the revised version is suitable for publication.

Please, find a list of changes at the end of this response.

A discussion about the novelty of the results presented in this manuscript arose in previous rounds of review. The authors acknowledge in their manuscript that "the crucial roles of non-equilibrium conditions and kinetic effects have been already investigated in molecular machines", including the role of allostery. Also, the fact that dissipative chemical systems can be interpreted as physical realizations of Maxwell's Demons has been shown in the literature cited in the manuscript. As the authors remarked, the novelty of the present contribution should be found in the kinetic model it proposes, which is new for the specific case of ABC transporters, to the best of my knowledge. Such a model differs from other minimalist models proposed in the literature for enzyme-driven transporter and leads to apparently different conclusions about what governs the steady-state substrate concentration on the two sides of a membrane [Acc. Chem. Res. 51, 2653 (2018); J. Am. Chem. Soc. 143, 5569 (2021)]. If the authors clarify why their model differs from other minimalist models, both in the choice of the minimal ingredients and in the predictions (see comment C1 for more details), I would confidently say, to the best of my knowledge, that that constitutes an element of novelty in the kinetic modelling of biologically relevant systems.

As the reviewer correctly points out, the analysis presented in our manuscript is similar to those carried out in the context of molecular machines, and also the result presented in Equation (3) of the main text shares similarities with previous results reported by the reviewer.

In the revised version of the manuscript, we highlight differences and analogies between our model and models employed to describe molecular motors. In particular, we comment on the crucial difference that the kinetic asymmetry factor, in our case, depend on some equilibrium constant. We also add the relevant literature mentioned by the reviewer.

This answer is detailed in the reply to Comment C1 (see below).

Unfortunately, I share some of the doubts of reviewer #2 concerning the rigorousness and generality of the Maxwell Demon analogy.

i) As the authors notice, "the Maxwell Demon does not perform any direct mechanical work on the system", meaning that the Demon and the system only exchange information. Would Maxwell's Demon analogy

hold strictly if the equilibrium ratio $[in]_{eq} / [out]_{eq} \neq 1$? In such a case, the Demon could alter the energy and not only the system's entropy, i.e., the Demon's action raises or lowers the substrate's energy by biasing it towards a region with a different standard chemical potential. In the language of Ref. 11 [Phys. Rev. X 4, 031015 (2014)], this would correspond, I think, to having an energy flow between the two subsystems. In my understanding, an autonomous Maxwell's Demon is such when the Demon (here, all the chemical reactions that do involve the binding and unbinding of the substrate?) and the system (here, the substrate "motion" across the barrier realized by some reactions?) only exchange information.

ii) Did the authors try to check if their definition of Maxwell's Demon is compatible with recent work on Characterizing autonomous Maxwell demons [Phys. Rev. E 103, 032118 (2021)]?

We thank the reviewer for going through the discussion with Reviewer #2.

Possibly, the best way to emphasize the role of the information exchange is to study $([in]/[out])/([in]^{eq}/[out]^{eq})$, which is equal to 1 when the Demon is not operating. This is the common settings also for standard Maxwell Demons, where the final distribution has to be compared with the equilibrium one. The latter, of course, can be affected by energy levels as the referee pointed out. ABC transporters only perform energetic discrimination at equilibrium, which is solely a consequence of the Boltzmann distribution. Notably, when we propose the information-theoretic approach, we always consider $([in]/[out])/([in]^{eq}/[out]^{eq})$, i.e., the displacement from equilibrium for which energy and information are needed. In the revised version, we tried to clarify further this point with a sentence in the information-theoretic part. Concerning the energy flows, of course at equilibrium they are absent. Out-of-equilibrium, there will be some energy flows together with the information exchange, but we do not think that this detail will affect the analogy with an autonomous Maxwell Demon. Moreover, even if the idealized model of an autonomous Maxwell Demon does not involve energy differences at equilibrium, real-world machines operating using information might exhibit a non-flat Boltzmann distribution (as in this case). Indeed, we have shown, using both the energy balance in the second part of the paper and Eq. (3), that all operations of an ABC transporters can be mapped into the ones performed by an autonomous MD. This hints at the fact that the presence of energy flows is immaterial for understanding how these molecular machines transduce information into a transmembrane chemical gradient.

For what concerns the categorization in Phys. Rev. E 103, 032118 (2021), we are unsure whether this particular setting detailed above in which there is energetic discrimination at equilibrium can fit in one of the proposed categories. Reviewer #2 proposed "non-equilibrium Maxwell Demon", but we do not think that their definition fit our model. Most importantly, we believe that the simple identification of ABC transporters as autonomous MD is enough to emphasize the idea that they use information to displace the transmembrane concentration gradient from its equilibrium value.

In the two previous paragraphs, I gave my frank opinion about this manuscript in the context of the previous discussion, which I couldn't avoid entering. Overall, my independent assessment of the work is positive, as I find it technically sound and interesting. However, I also think the authors can improve the impact of their work by addressing some "weaknesses" (to be understood as "missed opportunities" to make the paper better in connecting with previous literature). Concretely, I would be pleased to recommend the manuscript for publication after the authors have a chance to consider my two comments below.

We thank the reviewer the positive evaluation of our manuscript, for finding it technically sound and interesting, and for deeming it worth of publication in Communications Physics. Below, we present a detailed answer to all the points raised by the reviewer.

STRENGTHS:

S1) The authors did an admirable job going through the biological literature and distilling a new minimal model for ABC transporters, supported by experimental evidence and amenable to analytical analyses. Their model is a significant contribution that I consider worth publishing.

S2) The analysis of the model is insightful. Equation (3) neatly reveals three conditions that must be contemporarily satisfied to have transport. This theory-driven understanding vouches for the power and usefulness of physical modelling in identifying the fundamental elements of complex phenomena.

S3) The Maxwell's Demon analogy is well-motivated and leads to an elegant interpretation of each factor in equation (3) in terms of information-theoretical concepts. Although the analogy between Maxwell's Demons and chemical systems is not new, the authors' approach looks original in the context of ABC transporters.

We are very pleased by these reviewer's comments and glad to notice that they share our enthusiasm for having determined a minimal working model for ABC transporters which is both biologically and physically interpretable.

WEAKNESSES:

W1) In its current form, the manuscript fails to cite and contrast part of the results with previous similar results obtained in the context of chemical pumping, including other minimal models for enzymatic transporters. I see a missed opportunity to draw an insightful connection here, also in the light of seemingly contradictory results concerning previous models (see comment C1 for more details).

We thank the reviewer for giving us the opportunity to comment on this point.

As reported in the answer above and detailed below after Comment C1, in the revised version, we include a comparison between our model and the relevant literature in the context of molecular machines. In particular, we focus on the crucial difference in the kinetic asymmetric factor.

W2) In its current form, the manuscript fails to convincingly demonstrate that the Maxwell's Demon interpretation buys some new practical tools helping to characterize ABC transporters in practice better. This issue was put forward also by Referee #2, and the authors should have replied more convincingly. Let me be provocative: ABC Transporters are billion-year-old Maxwell Demons, and so what? There's room to address this critique by taking inspiration from previous literature (see comment C2 for more details).

We thank the referee for pointing out this very important aspect.

In the revised version, we strengthen the answer given to Reviewer #2, trying to bring novel and more compelling arguments. Moreover, we also add references to the relevant literature in this context reported in the reviewer's comment below.

A detailed answer is presented after Comment C2 (see below).

COMMENTS:

C1)

Molecular pumps have been analyzed by the molecular machines community, including kinetic models of ATP hydrolysis-driven transporters [Acc. Chem. Res. 2018, 51, 2653–2661; J. Am. Chem. Soc. 143, 5569 (2021)]. Equation (3) is very similar to the typical "pumping equations" derived in the context of molecular machines and motors following the same kind of analysis the authors perform (i.e., using Hill's spanning-tree method). In the context of molecular machines and motors, the analogue of the quantity in brackets in equation (3) is usually called the "kinetic asymmetry factor" and found to not depend on equilibrium constants but only on differences in transition states' energies [Chem. Soc. Rev., 2017, 46, 5491; Chem 6, 1952 (2020)]. The fact that, in the authors' model, the quantity in brackets in equation (3) does depend on (some) equilibrium constants is quite interesting, in my opinion, as it hints at practical ways to determine which model works best for a specific experiment. Therefore, I urge the authors to connect their ABC transporters with previous chemical pumping literature. In particular, it would be valuable to understand whether the dependency on some equilibrium constants in equation (3) is compatible with previous results (and arises, for instance, from model features) or if the authors think their result is somewhat in contrast with previous ones.

We thank the reviewer for this insightful comment. Indeed, it gave us the opportunity to dig into the literature the reviewer reported finding interesting similarities and differences with our model. The analysis performed within the context of molecular pumps is indeed analogous to the one presented in the manuscript, as also remarked above. The main structural difference within our model and those presented in the reported papers resides in the presence of the states T^* and T^*S . As highlighted in the manuscript, these two intermediate allosteric states are crucial to two main reasons. They provide a structural rationale for the substrate-induced change of the ATP hydrolysis rate, a biochemically compatible mechanism to have effective kinetic asymmetry, and they also are necessary to match experimental observations on ABC transporters. Indeed, the equilibrium constants appearing the Eq. (3) are those associated with the transitions from T to T^* and from TS to T^*S , and contribute to a new term that we identified as information feedback (in the analogy with a Maxwell Demon). This observation hints at the crucial role of allostery in this class of biochemical systems.

As the reviewer correctly points out, this difference between ABC transporters and molecular pumps arises from a biochemically-motivated model feature, and might be useful to perform model selection starting from observed behaviors.

In the revised version, we have added a discussion about these points, including also the relevant papers suggested by the reviewer.

C2)

Information-theoretical treatments of biologically relevant systems are typically used to define some performance quantifiers [for a review: *Front. Phys.* 11:1108357 (2023) doi: 10.3389/fphy.2023.1108357]. What are the authors gaining from calling the terms in equation (3) powering, measurement, feedback, and resetting other than a nice analogy? This question should and can be answered more convincingly by the authors, at least by acknowledging some examples where information-theoretical treatments have been used to quantify performance in biological systems [*Pnas* 119, 208083119 (2022); arXiv:2209.12084] or suggesting their strategy to put their information-theoretical understanding in practice.

We thank the reviewer for the opportunity to comment on this point.

First, the information-theoretic analysis has a three-fold theoretical advantage:

- i) to provide a quantitative connection between the terms in Eq. (3) and the operations of an MD, only qualitatively identified in the first part of the manuscript;
- ii) to hint at the existence of a mapping between autonomous and non-autonomous MD, at least in the case of ABC transporters, by looking at one specific cycle of operations;
- iii) to highlight an information-energy balance for ABC transporters that might be extended to other classes of biochemical transporters.

Although these points were mentioned in the previous version, we modified some sentences to clarify them in the revised version of the manuscript.

On a practical level, the estimation of the efficiency is usually done from the decomposition of the entropy production rate, as already mentioned in the main text. We also add the relevant papers reported by the reviewer that use, indeed, the same strategy as those already cited.

Furthermore, highlighting an energy balance that includes terms stemming from information processing features helps in quantifying energy transduction in our system. Indeed, part of the energy is used to sustain a transmembrane chemical gradient in excess over the equilibrium one, another part goes into dissipation, and another one is a cost to process information. This identification can be done for each cycle of operations, as shown in the paper. Moreover, these elements do not originate directly from the steady-state solution and their derivation required the presented additional analysis. Thus, tuning the transition rates or estimating them experimentally, is a way to respectively modulate or quantify how the system under investigation invests its energy budget, whether in the building of a gradient, dissipation, or information processing operations.

In the revised version, we also added some sentences highlighting another important related aspect. Describing the system starting at a coarser scale using the energy balance, Eq. (6), suggests that a more abstract information-theoretic depiction of biological systems, often advocated as the most appropriate beyond specific details, can be employed. This perspective indeed maintains its connection to molecular details, introducing them at a later stage, and might serve as a useful conceptual framework for understanding the complex biochemical circuitry of these systems.

List of changes in the revised version

- Page 4, line 162 – A discussion of similar approaches in the context of molecular pumps is added, as well as the relevant literature. We also motivated the presence of the equilibrium constants in Eq. (3), a crucial difference with previous models that can be used to perform model selection.
- Page 7, line 283 – We specified that the excess chemical potential across the membrane is already discounted of the intrinsic chemical potential difference. This is in analogy to Eq. (3), where the MD part multiplies the equilibrium ratio $[in]_{eq}/[out]_{eq}$.
- Page 7, line 312 – We clarified the definition of a bipartite network.
- Page 8, line 319 – We modified some sentences highlighting the advantages of our information-theoretic mapping (see points i-iii) in the answer above).
- Page 8, line 360 – We discussed another important (and general) aspect related to the proposed information-theoretic approach.

REVIEWERS' COMMENTS:

Reviewer #3 (Remarks to the Author):

The authors convincingly replied to my main points. I like the current form of the manuscript and recommend it for publication in Communications Physics.

I noticed that the authors wrote that they added references to the relevant literature in the context of my previous comment C2. However, I do not see the references I suggested cited in the revised version of the manuscript. If this is an error, the author should correct it. If this is a choice, I respect it. But I also stress that, in my opinion, the review [Front. Phys. 11:1108357 (2023) doi: 10.3389/fphy.2023.1108357] is an excellent reference for this paper as it deals with information-theoretical approaches in studying biological systems.

Reviewer #3

The authors convincingly replied to my main points. I like the current form of the manuscript and recommend it for publication in Communications Physics.

I noticed that the authors wrote that they added references to the relevant literature in the context of my previous comment C2. However, I do not see the references I suggested cited in the revised version of the manuscript. If this is an error, the author should correct it. If this is a choice, I respect it. But I also stress that, in my opinion, the review [Front. Phys. 11:1108357 (2023) doi: 10.3389/fphy.2023.1108357] is an excellent reference for this paper as it deals with information-theoretical approaches in studying biological systems.

We thank the reviewer for deeming our manuscript worth of publication and for the appreciation of the current form. We checked the revised version and the references were present, even if not highlighted in red. This might be the source of confusion. We apologize for it.

In the revised version, we added the reference suggested by the referee, as it was missing.